# Gradient Inversion Attacks Beyond SGD

**Guangnian Wan** [1]  **Gongfan Fang** [1]  **Xinyin Ma** [1]  **Xinchao Wang** [1]

## Abstract

Gradient Inversion Attack (GIA) poses a significant threat to federated learning, enabling adversaries to reconstruct private training data from the information shared during training. Prior research has predominantly focused on vanilla SGD, where the server or an eavesdropper can directly observe true gradients. In practical deployments, however, models may be trained with adaptive optimizers (e.g., Adam, RMSProp, and AdaGrad), for which the observable signal is not raw gradients but momentum-based parameter updates. This setting remains underexplored and undermines traditional gradient-matching strategies, which struggle to recover labels and images from non-gradient updates. To address this gap, this paper explores attacks tailored to modern adaptive optimizers. We present an analytical rule for recovering labels from optimizer updates and propose an update-matching objective that optimizes dummy inputs to reproduce the observed updates. The proposed approach is general and can be directly applied to various optimizers such as Adam, AdaGrad, and RMSProp. Furthermore, we find that, despite being introduced for adaptive optimizers, the proposed objective function also yields stronger attacks in the standard SGD setting. Experiments on datasets such as ImageNet and PACS highlight the effectiveness of our method over existing gradient matching techniques.

## 1. Introduction

Gradient inversion attacks (GIAs) (Zhu et al., 2019; Geiping et al., 2020; Huang et al., 2021) have emerged as a severe threat to the privacy of federated learning (FL) (McMahan

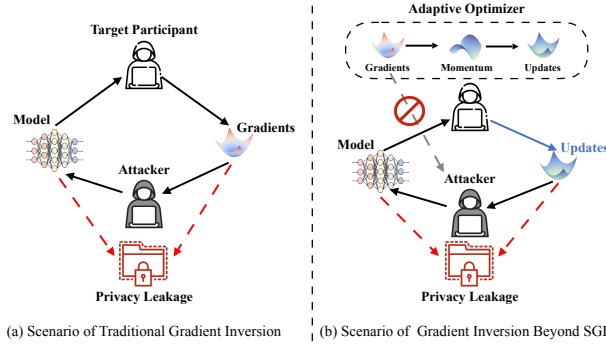

(a) Scenario of Traditional Gradient Inversion | (b) Scenario of Gradient Inversion Beyond SGD

*Figure 1.* Attack scenarios of traditional GIAs and gradient inversion beyond SGD. In traditional SGD-based scenarios, attackers have access to gradients. Beyond SGD, only model updates are available to attackers, rendering **raw gradients inaccessible**.

et al., 2017; Kairouz et al., 2021; Konečný, 2016). These attacks allow a curious participant, such as the FL server, to reconstruct clients' private data from the optimization signals shared during model training. Specifically, in each FL round, clients train the model locally on their private data and transmit updates (e.g., gradients) to a central server (Huang et al., 2021). An adversary with access to these signals can reconstruct clients' private data by generating inputs that best match the shared updates.

GIA originates from Deep Leakage from Gradients (DLG) (Zhu et al., 2019), which formulates an optimization procedure that reconstructs an input–label pair by matching gradients computed from dummy data to those uploaded by a victim client. Subsequent works further enhanced GIAs, achieving high-fidelity recovery of batch-level inputs by introducing improved gradient matching loss (Geiping et al., 2020), analytical label restoration methods (Zhao et al., 2020; Yin et al., 2021; Ma et al., 2023; Ye et al., 2024), additional regularization terms (Geiping et al., 2020; Yin et al., 2021; Fang et al., 2023), and Generative adversarial network (GAN) priors (Jeon et al., 2021; Li et al., 2022b; Fang et al., 2023; Goodfellow et al., 2020). Despite these advances, most GIA methods hinge on a key assumption: the attacker has direct access to gradients (Zhu et al., 2019; Geiping et al., 2020; Huang et al., 2021), as illustrated in the left part of Figure 1. In practice, however, FL clients may share optimizer-produced parameter updates rather than raw gradients, so this assumption holds primarily when the local

[1]Department of Electrical and Computer Engineering, National University of Singapore, Singapore, Singapore. Correspondence to: Xinchao Wang <xinchao@nus.edu.sg>.

*Proceedings of the 43rd International Conference on Machine Learning*, Seoul, South Korea. PMLR 306, 2026. Copyright 2026 by the author(s).

optimizer is standard SGD.

Currently, adaptive optimizers like Adam (Kingma & Ba, 2015) are widely used across modern architectures (Kunstner et al., 2023; Feng et al., 2025; Zelikman et al., 2022; Dosovitskiy et al., 2021), and recent studies have also incorporated adaptive optimizers into FL (Karimireddy et al., 2020). In this regime, the server typically observes optimizer-produced parameter updates rather than raw gradients, which undermines traditional GIA methods that rely on direct gradient access. Specifically, prior works typically optimize dummy inputs by minimizing a distance between dummy and true gradients (Zhu et al., 2019; Geiping et al., 2020; Huang et al., 2021) and leverage the mathematical properties of gradients to infer labels (Zhao et al., 2020; Yin et al., 2021; Ma et al., 2023; Ye et al., 2024). When raw gradients are not exposed, these objectives are undefined, and the associated label derivations no longer apply. Consequently, even without explicit privacy design, adaptive optimizers can substantially reduce the effectiveness of current GIA techniques.

In this paper, we propose a GIA method tailored to adaptive local optimizers, as shown in the right part of Figure 1. Our approach analytically extracts labels from model updates and generates inputs supervised by the true model updates through an optimization process (illustrated in Figure 2). Specifically, we derive element-wise equations for the final-layer gradients based on the adaptive optimizer's update rule. Because the per-element solutions can be non-unique, we treat them as candidate gradient values and recover a globally consistent gradient by enforcing constraints derived from the mathematical properties of final-layer gradients. Labels are then revealed from the recovered gradients. With these recovered labels, we initialize images from random noise and optimize them to minimize the distance between the model updates induced by the generated data and the observed updates from real data, thereby reconstructing private examples. Notably, although developed for adaptive optimizers, the proposed objective function with update-matching also improves reconstruction quality in the standard SGD setting when used in place of conventional gradient matching for an initialized FL model.

To evaluate the effectiveness of our method, we conduct experiments on the ImageNet (Deng et al., 2009) and PACS (Li et al., 2017) datasets, comparing our approach with IG (Geiping et al., 2020), GIAS (Jeon et al., 2021), GIFD (Fang et al., 2023), and HFGI (Ye et al., 2024) methods. In the adaptive-optimizer (update-only) setting, our method reconstructs images that closely resemble the originals, whereas the baselines produce hardly recognizable outputs. Quantitatively, our approach improves peak signal-to-noise ratio (PSNR) by up to 6.5 dB over these baselines. We also observe that the method generalizes effectively

across different optimizers such as Adam, RMSProp (Tieleman & Hinton, 2012), and AdaGrad (Duchi et al., 2011). Furthermore, even under standard SGD, where gradients are directly accessible, our approach outperforms the baselines by roughly 3.0 dB in PSNR.

**Contributions:** To the best of our knowledge, we are the first to show that GIAs can extend beyond FL algorithms that rely on local SGD. We propose the first GIA approach that generalizes effectively across various modern adaptive optimizers through an image optimization process and an analytical label-inference method. Additionally, the proposed update-matching objective also serves as a loss function that enables a stronger attack in the standard SGD setting.

## 2. Related Work

### 2.1. Gradient Inversion Attacks

Existing GIAs can generally be categorized into optimization-based and analysis-based methods.

**Optimization-based GIA.** Optimization-based GIA reconstructs private training data by optimizing dummy inputs so that their induced gradients match the observed ones (Du et al., 2025). Zhu et al. (2019) introduced the first optimization-based GIA, which iteratively optimizes a dummy input-label pair to minimize the distance between gradients from the true data and the dummy data. Zhao et al. (2020) enhanced this approach by analytically inferring the true labels in advance, optimizing only the input data thereafter. Geiping et al. (2020) advanced high-resolution image reconstruction on deep neural networks through a novel gradient-matching loss and a regularization term. Subsequent works enhanced reconstructions by incorporating additional priors, auxiliary statistics, or adaptive search strategies, including BN statistics (Yin et al., 2021), GAN priors (Goodfellow et al., 2020; Jeon et al., 2021; Fang et al., 2023), image feature priors (Usynin et al., 2023), adaptive neural architecture search (Yu et al., 2025), temporal-gradient statistics (Li et al., 2025), and language-guided semantic priors (Shan et al., 2025). Other studies have further broadened the scope of optimization-based attacks, involving vision transformers (Hatamizadeh et al., 2022; Lu et al., 2022), large-batch aggregated gradients (Li et al., 2022a), degraded gradients (Li et al., 2022b; Liu et al., 2025), multiple local training steps (Xu et al., 2022; Zhu et al., 2023), and duplicate-label batches (Ye et al., 2024).

**Analysis-based GIA.** Analysis-based GIA represents the other research direction. These attacks reconstruct training data by formulating and solving equations that link inputs with gradients (Aono et al., 2017; Zhu & Blaschko, 2021; Fowl et al., 2022; Kariyappa et al., 2023; Du et al., 2025;

Shi et al., 2025). Without the need to optimize generated data, these methods are generally more computationally efficient. However, they are often limited to specific network architectures (Ye et al., 2024; Fang et al., 2023) or may require a malicious server (Fowl et al., 2022; Shi et al., 2025). Despite their impressive performance, existing GIA methods have not considered the case where clients may use adaptive optimizers.

## 2.2. Adaptive local optimizer in Federated Learning

Adaptive optimization methods in FL can generally be classified into two categories: server-adaptive and client-adaptive methods (Jin et al., 2022). As server-adaptive methods do not change the threat model of existing GIAs, we focus on client-adaptive methods. In the literature, Reddi et al. (2021) proposed the first FL framework allowing both clients and the server to use adaptive optimizers. Liu et al. (2020) introduced momentum gradient descent as a local optimizer, requiring clients to send their optimizer states to the server for aggregation after each training round. Karimireddy et al. (2020) presented MIME, which incorporates adaptive optimizers (e.g., Adam) in local client training with shared optimizer states (e.g., momentum) across clients. Wang et al. (2021) suggested restarting local optimizer states at the start of each global training round. Sun et al. (2023) introduced client-level momentum to accelerate training and reduce heterogeneous overfitting. Wu et al. (2023) demonstrated that independently combining adaptive optimizers and local training on each client can cause divergence and proposed a momentum-based variance reduction method for local adaptive optimization. Additionally, Lewis et al. (2024) proposed a privacy-preserving approach using private adaptive optimizers, concealing the initial state of the adaptive local optimizer from the server through secure aggregation (Bonawitz et al., 2016). However, secure aggregation techniques can impose substantial communication overhead (Lewis et al., 2024; Ye et al., 2024), and an attacker may even evade the secure aggregation protocol (Pasquini et al., 2022).

## 3. Method

In this section, we first introduce the problem formulation and the threat model, by which we explain the main challenges in previous paradigms. We then present our method, which enables data reconstruction beyond the SGD setup, incorporating image optimization and label recovery.

**Problem Formulation.** Consider a model $f_\theta$ with parameters $\theta$ for image classification tasks with cross-entropy loss $\ell(\cdot)$, and a batch of training data pairs $(x, y) = \bigcup_{i=1}^{B}\{(x_i, y_i)\}$ with image $x_i$ and label $y_i$, the gradients can be calculated as $\nabla\theta = \frac{1}{B}\sum_{i=1}^{B}\nabla_\theta\ell(f_\theta(x_i), y_i)$. In

a conventional SGD-based FL system, once the attacker obtains the gradients $\nabla\theta$, it typically reconstructs the private data through two stages. First, it analytically infers the labels $\hat{y} \in \{0, 1\}^{B \times L}$, where $L$ denotes the number of classes, from $\nabla\theta$ by exploiting the mathematical properties of the final-layer gradients (Zhao et al., 2020; Yin et al., 2021; Ma et al., 2023; Ye et al., 2024). Given the recovered labels, the attacker then optimizes a dummy image tensor $\hat{x} \in \mathbb{R}^{B \times H \times W \times C}$ ($B$, $H$, $W$, and $C$ denote the batch size, height, width, and number of channels, respectively) by solving the following gradient-matching objective:

$$\hat{\mathbf{x}}^* = \arg\min_{\hat{\mathbf{x}}} \mathcal{D}\left(\nabla\hat{\theta}, \nabla\theta\right) + \alpha\mathcal{P}(\hat{\mathbf{x}}), \qquad (1)$$

where $\nabla\hat{\theta} = \frac{1}{B}\sum_{i=1}^{B}\nabla\ell(f_\theta(\hat{\mathbf{x}}_i), \hat{\mathbf{y}}_i)$ is the average gradient computed from the dummy inputs and the analytically inferred labels, $\mathcal{D}(\cdot)$ measures the distance between gradients, $\mathcal{P}(\hat{\mathbf{x}})$ is an image regularization term (e.g., total variation (Geiping et al., 2020)), and $\alpha$ is the corresponding coefficient. However, when FL clients use adaptive optimizers for local training, model updates are computed as $\theta \leftarrow \theta - \eta\mathcal{U}(\nabla\theta, s)$, where $\eta$ is the learning rate, $\mathcal{U}(\cdot)$ denotes the optimizer-specific update rule, and $s$ is the initial state of the optimizer (e.g., momentum). As a result, attackers no longer have direct access to the raw gradients, hindering prior reconstruction methods that rely on them.

**Threat Model.** We investigate the scenario in which an honest but curious server attempts to reveal clients' private data from their uploaded updates. The server can receive model updates from any client. Since existing adaptive FL algorithms typically unify the optimizer state across clients at the beginning of each local training round (Karimireddy et al., 2020; Wu et al., 2023; Sun et al., 2023), we assume the adversary has access to the initial state (e.g., momentum) of each local optimizer but not to the state after training. For instance, the server can access the initial values of the first- and second-order momentum in the Adam optimizer, but not their values after training.

**Challenges in Previous Paradigms.** The main challenges brought by the absence of raw gradients stem from two aspects: **(1) Objective Function.** Previous works optimize generated images guided by the distance between the generated gradients and true gradients, as shown in Equation 1. However, the absence of raw gradient information prevents the calculation of these objective functions. **(2) Label Restoration.** Prior methods typically exploit the mathematical properties of gradients to derive label information. For a simple illustration, when the training data is a single image, the ground truth label can be accurately inferred through (Zhao et al., 2020):

$$c = i, \quad \text{s.t.} \quad \nabla\mathbf{W}_{\text{FC}}^{i}{}^{\top} \cdot \nabla\mathbf{W}_{\text{FC}}^{j} \leq 0, \quad \forall j \neq i, \qquad (2)$$

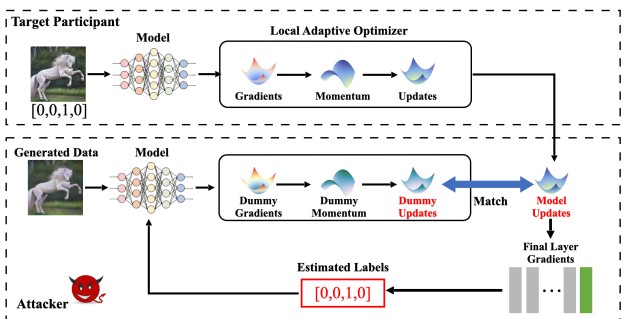

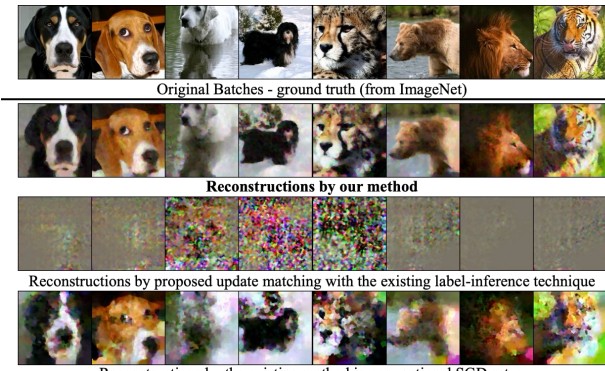

Original Batches - ground truth (from ImageNet)

Reconstructions by our method

Reconstructions by proposed update matching with the existing label-inference technique

Reconstructions by the existing method in conventional SGD setup

*Figure 2.* Overview of our proposed method. The generated images are optimized to minimize the distance between the dummy updates and the shared updates from the target participant, with labels analytically inferred. To reveal label information, the attacker first recovers the final-layer gradients from the model updates, then extracts the labels from the recovered gradients.

*Figure 3.* Reconstruction results of (1) our method incorporating image optimization using Equation 3 and our proposed analytical label recovery, (2) the proposed update-matching objective with the existing label-inference technique, (3) the existing method in the conventional SGD setup.

where $\nabla \mathbf{W}_{\text{FC}}^i$ is the final layer gradient vector w.r.t the weights connected to the $i_{th}$ logit. However, without access to raw gradients, these types of derivations become infeasible, hindering accurate label inference of previous methods.

To address these challenges, we iteratively optimize the generated images to search for the set of inputs that best match the true model updates, with one-hot labels analytically derived solely from the model updates. An overview of our method is shown in Figure 2.

### 3.1. Image Optimization

We reconstruct the images by iterative optimization using the following objective function:

$$\hat{\mathbf{x}}^* = \arg\min_{\hat{\mathbf{x}}} \left( 1 - \frac{\langle \mathcal{U}(\nabla\hat{\theta}, s), \mathcal{U}(\nabla\theta, s) \rangle}{\|\mathcal{U}(\nabla\hat{\theta}, s)\| \|\mathcal{U}(\nabla\theta, s)\|} + \alpha \mathcal{P}(\hat{\mathbf{x}}) \right). \tag{3}$$

Since Adam is one of the most widely used adaptive optimizers, we use it as an example for the following analysis, which can be easily generalized to other adaptive optimizers. The computation rule of Adam can be denoted as:

$$\mathcal{U}(\nabla\theta, s) = \hat{m}_t(\nabla\theta) / \left( \sqrt{\hat{v}_t(\nabla\theta)} + \epsilon \right), \tag{4}$$

where $\hat{m}_t(\nabla\theta) = \frac{\beta_1}{1-\beta_1^t} m_{t-1} + \frac{1-\beta_1}{1-\beta_1^t}\nabla\theta$, and $\hat{v}_t(\nabla\theta) = \frac{\beta_2}{1-\beta_2^t} v_{t-1} + \frac{1-\beta_2}{1-\beta_2^t}\nabla\theta^2$. $m_t$ and $v_t$ represent the first-order and second-order momentum; $\beta_1$ and $\beta_2$ are the averaging factors; and $\epsilon$ is the numerical stability term. Specifically, we start with randomly initialized dummy inputs to compute dummy gradients. Using the initial state (i.e., initial momentum values) of the local optimizer and these dummy gradients, we calculate the dummy momentum, enabling us to derive the dummy updates. The loss is then computed by measuring the negative cosine similarity between the

dummy updates and the true model updates. We incorporate total variation (Geiping et al., 2020) as the regularization term. Additionally, we apply random drop (Ye et al., 2024) to the model updates for each layer at each optimization step that requires matching.

With the image-optimization objective, a straightforward attempt is to combine it with existing analytical label-inference methods. To evaluate the feasibility of this approach, we conduct attack experiments on ImageNet with a randomly initialized ResNet-18 (He et al., 2016) model, using the label-inference strategies adopted in classical gradient inversion attacks (Zhao et al., 2020; Geiping et al., 2020). The reconstruction results of this approach are shown in the third row of Figure 3. The barely recognizable images show that directly pairing prior label-inference methods with update matching is insufficient for faithful reconstruction. We therefore introduce an analytical label-recovery strategy for the adaptive-optimizer setting, and couple it with the update-matching objective. As shown in the second row of Figure 3, this combination achieves high-quality reconstructions.

Additionally, we compare the reconstructions from our attack in the Adam setting and those from the previous method IG (Geiping et al., 2020) in the traditional SGD setup (fourth row in Figure 3). The results demonstrate that our method produces even higher-fidelity images than those obtained by the previous approach using gradient matching. In this case, since both the labels recovered by our method and the previous method are accurate, the advantage of our method stems from our new objective function in Equation 3. Notably, although FL clients do not compute or upload adaptive updates in traditional SGD settings, an attacker can actively construct adaptive updates to utilize the loss function in Equation 3. For instance, an attacker can estimate an optimizer state $s$ using the historical gradient information it received. Given this state, upon receiving new gradients

$\nabla\theta$, the attacker can compute $\mathcal{U}(\nabla\theta, s)$, thereby computing Equation 3. Consequently, our work not only extends gradient inversion risks to adaptive optimizers but also can serve as a stronger attack in a traditional SGD setting, which will be shown in Section 4.1 and Section 4.4, respectively.

### 3.2. Label Recovery

Previous works have demonstrated that ground-truth labels can be accurately inferred from the final-layer gradients (Zhao et al., 2020; Yin et al., 2021; Ma et al., 2023; Ye et al., 2024). Inspired by this, we recover the gradients of the final layer from the model updates, and then extract the labels from the recovered gradients. Based on Equation 4, we can formulate an equation:

$$\mathcal{U}(\nabla\theta, s) = \frac{\frac{\beta_1}{1-\beta_1^t} m_{t-1} + \frac{1-\beta_1}{1-\beta_1^t} \nabla\theta}{\left(\frac{\beta_2}{1-\beta_2^t} v_{t-1} + \frac{1-\beta_2}{1-\beta_2^t} \nabla\theta^2\right)^{\frac{1}{2}} + \epsilon}. \quad (5)$$

Since $\epsilon$ is very small, it can typically be ignored. Additionally, attackers can legally obtain $\mathcal{U}(\nabla\theta, s)$, $m_{t-1}$, $v_{t-1}$, $\beta_1$, and $\beta_2$. Therefore, each element of $\nabla\theta$ in Equation 5 can be solved as a quadratic equation, yielding one or two solutions. Then the key is to establish sufficient constraints to identify the true gradients from these potential values.

Since solving Equation 5 yields a unique solution for some gradient elements, these can be fixed directly without additional constraints. To investigate this, we empirically tracked the number of elements with two candidate solutions over training steps. We observe that this count drops rapidly as training progresses. That is, the number of uniquely solvable elements rises quickly. This suggests that, outside the initial phase, only a small fraction of elements require extra constraints to be determined after solving Equation 5, which simplifies gradient recovery. Guided by this observation, we first focus on the post-initial phase and then move on to the initial phase. Detailed results are provided in the Appendix.

**Phase After Initial Training Stage.** In this case, we observe that elements with two candidate solutions tend to cluster in specific rows, with indices corresponding to labels present in the training batch. Guided by this observation, we establish constraints from the perspective of columns.

For a fully connected layer, we have $o = Wz + b$, where $o \in \mathbb{R}^{q \times 1}$, $W \in \mathbb{R}^{q \times p}$, $z \in \mathbb{R}^{p \times 1}$, and $b \in \mathbb{R}^{q \times 1}$. Here, $o, W, z, b, p, q$ represent the output, weights, input, bias, input dimension, and output dimension of the layer, respectively. In this context, for any row index $r \in [0, q)$, the

following holds:

$$\overline{\nabla W_r} = \frac{1}{B}\sum_{i=1}^{B} \nabla W_{i,r} = \frac{1}{B}\sum_{i=1}^{B} \frac{\partial \ell(x_i, y_i)}{\partial o_{i,r}(x_i)} z_i^T$$

$$\overline{\nabla b_r} = \frac{1}{B}\sum_{i=1}^{B} \nabla b_{i,r} = \frac{1}{B}\sum_{i=1}^{B} \frac{\partial \ell(x_i, y_i)}{\partial o_{i,r}(x_i)}. \quad (6)$$

Based on Equation 6, the true gradients can be identified using the following constraints:

$$\sum_r \overline{\nabla b_r} = \frac{1}{B}\sum_{i=1}^{B}\sum_r \frac{\partial l(x_i, y_i)}{\partial b_r} = 0$$

$$\sum_r \overline{\nabla W_{r,c}} = \frac{1}{B}\sum_{i=1}^{B}\sum_r \frac{\partial l(x_i, y_i)}{\partial W_{r,c}} = 0, \quad (7)$$

where $c$ can be any column index. Specifically, we identify the true gradients of each column by selecting the unique combination of candidate values that ensures the sum of gradients in the column is zero. Next, using the recovered final-layer gradients, the labels can be extracted by leveraging existing methods (Ma et al., 2023; Ye et al., 2024).

**Initial Training Phase.** In the initial training phase, each column contains numerous elements with two candidate values. This makes it challenging to efficiently identify the true values of each gradient element using only the constraints from Equation 7. To address this, we introduce additional constraints based on the characteristics of models during the initial training phase. We first divide the training batch $(x, y)$ into $T$ subsets according to labels, specifically, $(x, y) = \{\mathbb{B}_1, \ldots, \mathbb{B}_T\}$. Next, we introduce an approximation (Ma et al., 2023) and an assumption (Yin et al., 2021).

**Approximation 3.1** (Inter-class Low Entanglement of Gradient Contributions). For a model in the initial training stage, the batch-averaged gradient row at index $r$ is mainly from samples of $r - th$ class in the training batch. Specifically, if $\mathbb{B}_r \neq \emptyset$, we have:

$$\overline{\nabla b_r} = \frac{1}{B}\sum_{j=1}^{T} |\mathbb{B}_j| \overline{\nabla b_{r,\mathbb{B}_j}} \approx \frac{|\mathbb{B}_r|}{B} \overline{\nabla b_{r,\mathbb{B}_r}}$$

$$\overline{\nabla W_r} = \frac{1}{B}\sum_{j=1}^{T} |\mathbb{B}_j| \overline{\nabla W_{r,\mathbb{B}_j}} \approx \frac{|\mathbb{B}_r|}{B} \overline{\nabla W_{r,\mathbb{B}_r}}. \quad (8)$$

**Assumption 3.1** (Non-negative Activation Function). The input of the final fully connected layer is non-negative, i.e., $z_{n,i} >= 0$, where $z_{n,i}$ is the $n - th$ element of $z_i$.

Assumption 3.1 holds if the preceding layer has a commonly used activation function such as ReLU (Glorot et al., 2011) or Sigmoid. Combining Approximation 3.1, Assumption 3.1, and Equation 7, for any $\overline{\nabla W_{r,c}} \neq 0$ and $\overline{\nabla b_r} \neq 0$,

*Table 1.* Quantitative comparison of our method with baselines on images of the ImageNet and PACS datasets. We calculate the average value of metrics on reconstructed images. ↑: the higher the metric, the better the performance. ↓: the lower the metric, the better the performance.

| Metric | ImageNet | | | | | PACS | | | | |
|---|---|---|---|---|---|---|---|---|---|---|
| | IG | GIAS | GIFD | HFGI | **Ours** | IG | GIAS | GIFD | HFGI | **Ours** |
| MSE↓ | 0.0523 | 0.0806 | 0.0778 | 0.0519 | **0.0099** | 0.0697 | 0.1067 | 0.1069 | 0.0611 | **0.0142** |
| PSNR↑ | 12.9956 | 11.1840 | 11.3225 | 13.3472 | **20.5985** | 11.6007 | 9.9322 | 9.9342 | 12.4712 | **19.0078** |
| SSIM↑ | 0.1730 | 0.1524 | 0.1579 | 0.2507 | **0.5026** | 0.1705 | 0.1604 | 0.1700 | 0.3259 | **0.5192** |
| LPIPS (AlexNet)↓ | 0.9008 | 0.8737 | 0.8122 | 0.8328 | **0.4230** | 0.8625 | 0.8796 | 0.8308 | 0.7392 | **0.3919** |
| LPIPS (VGG)↓ | 0.8284 | 0.8044 | 0.8019 | 0.8095 | **0.5324** | 0.8391 | 0.8348 | 0.8273 | 0.7509 | **0.5202** |

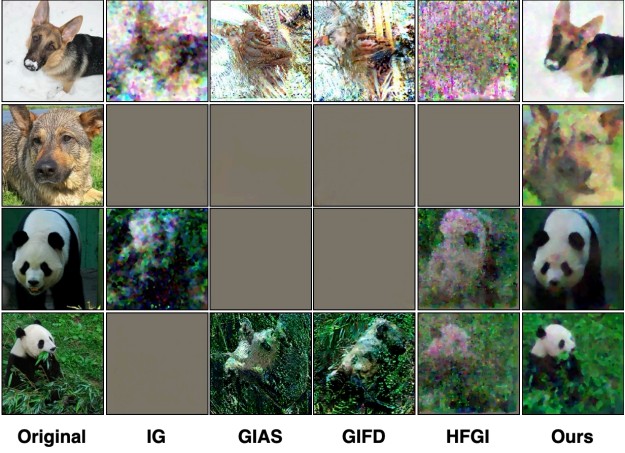
*(a)* ImageNet Attack with Raw Gradient Inaccessible

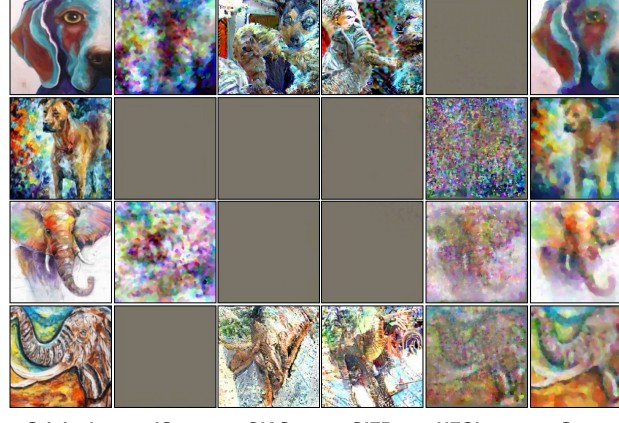
*(b)* PACS Attack with Raw Gradient Inaccessible

*Figure 4.* Visual comparison of our method with baselines on images of the ImageNet and PACS datasets with $B = 4$.

the gradients of the last layer satisfy the following condition:

$$\begin{cases} \sum_r \overline{\nabla b_r} = 0, \\ \sum_r \overline{\nabla W_{r,c}} = 0, \\ |B|\overline{\nabla b_r} >= -|\mathbb{B}_r| \\ \overline{\nabla W_{r,c}} < 0 \Rightarrow \mathbb{B}_r \neq \emptyset \\ (|B|\overline{\nabla b_r} \approx -|\mathbb{B}_r|) \wedge (\overline{\nabla b_r} < 0) \iff \mathbb{B}_r \neq \emptyset. \end{cases}$$

Based on the above properties together with the candidate gradient values, label recovery can be formulated as selecting gradients and values of $|\mathbb{B}_r|$ that satisfy the above constraints, with $|\mathbb{B}_r|$ corresponding to the number of data points in the batch with label $r$. For each label index $r$, a candidate value of $\overline{\nabla b_r}$ suggests possible values of $|\mathbb{B}_r|$ through the relation $|B|\overline{\nabla b_r} \approx -|\mathbb{B}_r|$, with the sign requirement $\overline{\nabla b_r} < 0$ when $\mathbb{B}_r \neq \emptyset$ and the bound $|B|\overline{\nabla b_r} \geq -|\mathbb{B}_r|$. These possible values are further constrained by the weight-gradient candidates: for any pair $(r, c)$, if all retained candidate values of $\overline{\nabla W_{r,c}}$ are negative, then $\overline{\nabla W_{r,c}} < 0 \Rightarrow \mathbb{B}_r \neq \emptyset$ requires $|\mathbb{B}_r| > 0$. We then choose the gradients and the values of $|\mathbb{B}_r|$ that jointly satisfy these local constraints and the global consistency constraints $\sum_r |\mathbb{B}_r| = |B|$ and $\sum_r \overline{\nabla b_r} = 0$. The resulting values $\{|\mathbb{B}_r|\}_r$ recover the labels of the entire training batch.

Although the above derivation relies on Assumption 3.1, the label recovery procedure can be extended to activation functions whose outputs can be negative but are lower-bounded, such as GELU or SiLU. To this end, we replace the rule that treats any negative $\overline{\nabla W_{r,c}}$ as evidence for label presence with an empirical thresholded rule, while keeping the rest of the label recovery procedure unchanged. Specifically, the original criterion that uses $\overline{\nabla W_{r,c}} < 0$ to indicate $\mathbb{B}_r \neq \emptyset$ is replaced by $\overline{\nabla W_{r,c}} < -\tau$, where $\tau > 0$ is a small threshold. The rationale is that, although such activation functions may produce negative values, their negative ranges are lower-bounded. Thus, when class $r$ is absent, negative $\overline{\nabla W_{r,c}}$ may arise because of negative activation values, but such values are typically much smaller in magnitude compared to the case where $\mathbb{B}_r \neq \emptyset$. Detailed proofs for the results in this section are provided in Appendix A.

## 4. Experiments

To evaluate the effectiveness of our method in reconstructing private data from model updates, we conduct attack experiments on image classification tasks using the ImageNet ILSVRC 2012 (Deng et al., 2009) and PACS (Li

*Table 2.* Quantitative results of our method and IG with different adaptive optimizers.

| Optimizer | Method | Metric | | | |
|---|---|---|---|---|---|
| | | PSNR↑ | LPIPS↓ | SSIM↑ | MSE↓ |
| Momentum | IG | 10.6764 | 0.7759 | 0.2891 | 0.0881 |
| | Ours | 17.0135 | 0.5577 | 0.4128 | 0.0226 |
| AdaGrad | IG | 9.6361 | 0.7839 | 0.2847 | 0.1098 |
| | Ours | 18.7233 | 0.4782 | 0.4265 | 0.0154 |
| RMSProp | IG | 11.8515 | 0.7676 | 0.3033 | 0.0667 |
| | Ours | 20.4626 | 0.4590 | 0.4993 | 0.0100 |

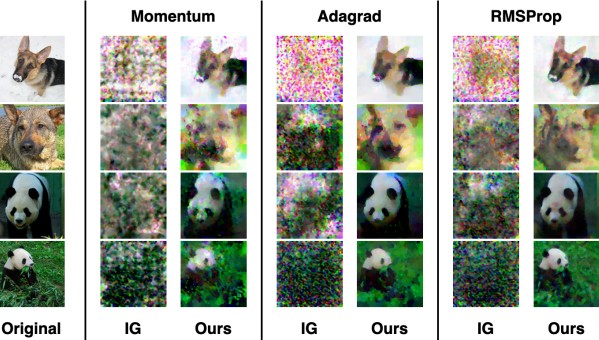

*Figure 5.* Reconstructions with different adaptive optimizers.

et al., 2017) datasets with 224×224-pixel images. We use ResNet-18 (He et al., 2016) as the FL model and Adam as the default local adaptive optimizer, with additional experiments conducted using other adaptive optimizers, such as RMSProp and AdaGrad. We set $B = 4$ by default and also experimented with different batch sizes to assess the impact of batch size on reconstruction quality.

**Implementation Details.** In our attack method, we use Adam to optimize the generated inputs. The learning rate is initialized to $0.1$ and decayed using a step schedule. For each attack experiment, we optimize the batch for 24000 iterations, randomly dropping 30% of the model-update entries (Ye et al., 2024) for each layer at each step. The coefficient of $\alpha$ in Equation 3 is set to $5 \times 10^{-3}$ for ImageNet and $10^{-2}$ for PACS. To simulate the initial state of the local optimizer, we trained the model from scratch and recorded the optimizer state at different training stages. Additional details are provided in Appendix B.

Besides (1) visual comparison, we report the following quantitative metrics to assess reconstruction quality: (2) Mean Squared Error (MSE), (3) Peak Signal-to-Noise Ratio (PSNR), (4) Structural Similarity Index Measure (SSIM), and (5) Learned Perceptual Image Patch Similarity (LPIPS) (Zhang et al., 2018) computed with AlexNet (Krizhevsky et al., 2012) and VGG network (Simonyan & Zisserman, 2015).

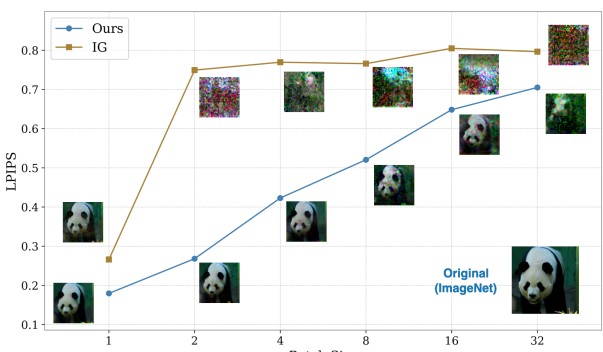

*Figure 6.* The impact of different batch sizes on LPIPS value.

### 4.1. Comparison with Gradient-based Attacks

We compare our proposed method with the following state-of-the-art baselines: (1) IG by Geiping et al. (2020), (2) GIAS by Jeon et al. (2021), (3) GIFD by Fang et al. (2023), and (4) HFGI by (Ye et al., 2024). These baselines cover both classical and recent GIAs, including methods with and without generative priors. Since the gradient-matching loss of the baselines cannot be calculated, we adjust their objectives from gradient matching to model-update matching. For GIAS and GIFD, we use a pre-trained BigGAN (Brock et al., 2019). For IG, the original label inference rule is not applicable when $B > 1$, so we use the extension of this rule introduced by Yin et al. (2021) in this setting. Following previous works (Zhu et al., 2019; Fang et al., 2023; Geiping et al., 2020), we set the model parameters to be randomly initialized by default. We use the optimizer state after the first step of the second training epoch to simulate the target client's optimizer state. We also conduct experiments to examine the impact of model parameters and optimizer state at different training stages, with results provided in Appendix C. Since FL clients may not transmit BN statistics computed on their private data, all experiments are conducted without the BN prior proposed by Yin et al. (2021).

We present a visual and quantitative comparison of our method with baselines in Figure 4 and Table 1. Overall, our method outperforms baseline GIAs for data reconstruction when only model updates from Adam are available. As shown in Figure 4, for both the ImageNet and PACS datasets, baseline methods struggle to generate meaningful content, while our method produces images closely resembling the real ones. In the quantitative comparison shown in Table 1, our method surpasses the best baseline results by nearly 6.5 dB, 0.20, 0.35, and 0.23 in terms of PSNR, SSIM, LPIPS (AlexNet), and LPIPS (VGG), respectively.

### 4.2. Different adaptive optimizers

In addition to Adam, we evaluate our method using other adaptive optimizers, including RMSProp and AdaGrad. Re-

*Table 3.* Quantitative comparison of our method with baselines on images of the ImageNet and PACS datasets under the traditional SGD setting, where **raw gradients are directly accessible**.

| Metric | ImageNet | | | | | PACS | | | | |
|---|---|---|---|---|---|---|---|---|---|---|
| | IG | GIAS | GIFD | HFGI | **Ours** | IG | GIAS | GIFD | HFGI | **Ours** |
| MSE↓ | 0.0430 | 0.0398 | 0.0321 | 0.0086 | **0.0046** | 0.0271 | 0.0250 | 0.0373 | 0.0150 | **0.0034** |
| PSNR↑ | 14.3207 | 14.0083 | 15.0285 | 20.7997 | **23.9195** | 17.3832 | 16.3876 | 14.8280 | 19.2953 | **24.8643** |
| SSIM↑ | 0.3782 | 0.3666 | 0.3923 | 0.5211 | **0.5744** | 0.4662 | 0.4631 | 0.4249 | 0.5111 | **0.6303** |
| LPIPS (AlexNet)↓ | 0.7293 | 0.6558 | 0.6094 | 0.4826 | **0.3886** | 0.5600 | 0.4690 | 0.5126 | 0.4490 | **0.2838** |
| LPIPS (VGG)↓ | 0.7161 | 0.6746 | 0.6436 | 0.5458 | **0.4681** | 0.5938 | 0.5504 | 0.5874 | 0.5410 | **0.4049** |

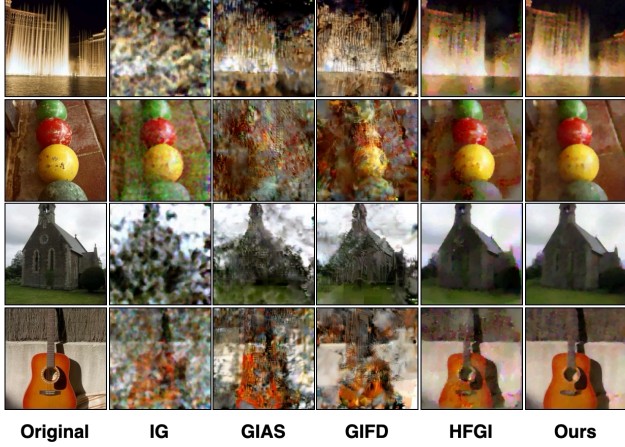 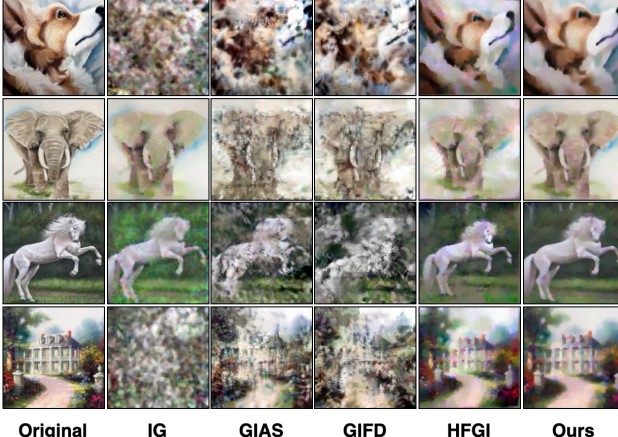

*(a)* ImageNet Attack with Raw Gradient Accessible       *(b)* PACS Attack with Raw Gradient Accessible

*Figure 7.* Visual comparison of our method with baselines on images of the ImageNet and PACS datasets under the traditional SGD setting, where **raw gradients are directly accessible**. In this case, the proposed method still outperforms existing methods.

sults with SGD with momentum are also reported. Since SGD with momentum only applies a linear transformation to the original gradients, these gradients can be directly recovered through a simple linear transformation of the model updates, given the initial state of the optimizer. Accordingly, we match the recovered gradients for this optimizer. For all other cases, we match the model updates. The visual and quantitative results of IG and our method are shown in Figure 5 and Table 2, respectively. As shown, our method generalizes well across different optimizers. It reconstructs images that are visually close to the real ones across the three optimizers, while IG struggles to produce high-quality reconstructions under these optimizer settings.

### 4.3. Effect of Batch Size

We then conduct experiments on ImageNet to evaluate the effect of batch size on recovered images. The average LPIPS values, computed using AlexNet across images in the batches from our method and IG (Geiping et al., 2020), are presented in Figure 6. The LPIPS values increase with larger batch sizes, indicating a decline in reconstruction quality for both our method and IG. However, the reconstructions generated by our method remain similar to the original images

up to $B = 16$. In contrast, the IG method achieves high-fidelity reconstructions only at $B = 1$, with reconstructions becoming unrecognizable as batch size increases.

### 4.4. Comparison in Traditional SGD Setup

To further validate that the proposed objective improves reconstruction quality, we evaluate in the standard SGD setting, where raw gradients are accessible. Specifically, we provide these gradients to all baselines so that they can use their native gradient-matching objectives, whereas our method deliberately retains the update-matching objective in Equation 3 for Adam update matching.

In the standard SGD setting, our method still achieves superior performance. As shown in Table 3 and Figure 7, IG's performance varies with image content—some images are reconstructed well, whereas others degenerate into noise. GIAS and GIFD, while effective on 64×64 images (Fang et al., 2023; Jeon et al., 2021), often produce blurry results at 224×224 resolution. HFGI generates visually similar images to the real ones, yet still underperforms compared to our method in both qualitative and quantitative evaluations. Quantitatively, our method outperforms the best baseline

*Table 4.* Quantitative results of ablation study on ImageNet.

| Method | Metric | | | |
|---|---|---|---|---|
| | PSNR↑ | LPIPS↓ | SSIM↑ | MSE↓ |
| w/o Label | 15.3358 | 0.7175 | 0.3104 | 0.0352 |
| w/o Objective | 11.4221 | 0.8413 | 0.2471 | 0.0776 |
| Ours | **19.6252** | **0.4614** | **0.4981** | **0.0125** |

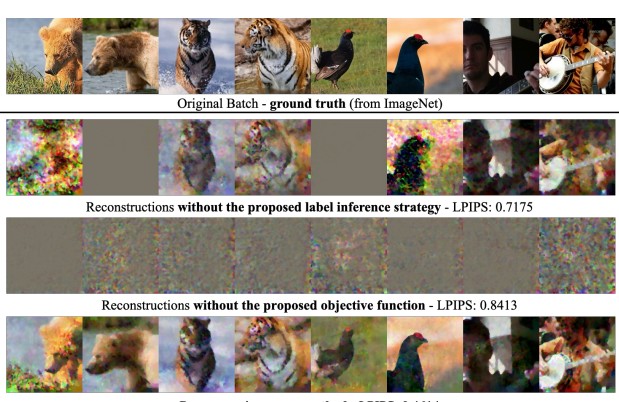

Original Batch - **ground truth** (from ImageNet)

Reconstructions **without the proposed label inference strategy** - LPIPS: 0.7175

Reconstructions **without the proposed objective function** - LPIPS: 0.8413

Reconstructions - **our method** - LPIPS: 0.4614

*Figure 8.* Visual results of the ablation study on ImageNet.

by approximately 3.0 dB in PSNR, 0.05 in SSIM, 0.09 in LPIPS (AlexNet), and 0.08 in LPIPS (VGG).

## 4.5. Ablation Studies

We conduct ablation experiments on ImageNet to assess the contribution of each component via two ablated variants: w/o Label and w/o Objective. In the w/o Label variant, we replace our proposed label-inference method with the strategy introduced by Ye et al. (2024), which has been shown to be effective under the conventional SGD setting. In the w/o Objective variant, we replace the proposed objective function with a gradient-matching objective that minimizes the distance between the gradients computed from the generated data and the observed model updates. The results in Table 4 and Figure 8 show that both the proposed objective function and label-inference method are critical for high-quality reconstructions.

## 5. Conclusion

In this paper, we introduce an approach that can reconstruct private training data in FL from model updates produced by adaptive optimizers. This relaxes a common assumption in prior GIAs that limits attacks to SGD-based FL. The attack builds its success on an optimization-based image-generating technique alongside an analytical label-recovery method. Our experimental results on two image classification datasets demonstrate the effectiveness of our method across various adaptive optimizers and its superiority over

existing attacks. We hope this work contributes to the development of more secure FL systems in the future.

## Acknowledgement

This project is supported by the National Research Foundation, Singapore, and Cyber Security Agency of Singapore under its National Cybersecurity R&D Programme and CyberSG R&D Cyber Research Programme Office (Award: CRPO-GC1-NTU-002).

## Impact Statement

This paper presents work whose goal is to advance the field of Machine Learning. There are many potential societal consequences of our work, none of which we feel must be specifically highlighted here.

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

# A. Detailed Proof

## A.1. Proof of Equation 7

For the final layer in a model with multi-class classification tasks. We have $o = Wz + b$, where $o, W, z, b$ represent the output, weights, input, and bias of this layer, respectively. For any data index $i$ and row $r$ in $W$, we have:

$$\frac{\partial \ell(x_i, y_i)}{\partial W_r} = \frac{\partial \ell(x_i, y_i)}{\partial o_{i,r}} z_i^T, \tag{9}$$

and for any row $r$ in $b$, we have:

$$\frac{\partial \ell(x_i, y_i)}{\partial b_r} = \frac{\partial \ell(x_i, y_i)}{\partial o_{i,r}}. \tag{10}$$

When the output $o_i$ from the final layer is passed through softmax and cross-entropy, the gradients of $o_{i,r}$ for any given input $x_i$ with hard label $y_i$ are:

$$\frac{\partial l(x_i, y_i)}{\partial o_{i,r}} = \begin{cases} p_{i,r} - 1 & \text{if } r = y_i \\ p_{i,r}, & \text{if } r \neq y_i \end{cases}, \tag{11}$$

where $p_i$ is the predicted probability for the $i_{th}$ class after applying softmax.
Combining Equation 9, Equation 10, and Equation 11, we obtain:

$$\frac{\partial l(x_i, y_i)}{\partial W_r} = \begin{cases} (p_{i,r} - 1) z_i^T & \text{if } r = y_i \\ p_{i,r} z_i^T, & \text{if } r \neq y_i \end{cases}$$
$$\frac{\partial l(x_i, y_i)}{\partial b_r} = \begin{cases} p_{i,r} - 1 & \text{if } r = y_i \\ p_{i,r}, & \text{if } r \neq y_i \end{cases}. \tag{12}$$

For any given data point $x_i$ and column index $c$, since $\sum_r p_{i,r} = 1$, we have:

$$\sum_r \frac{\partial l(x_i, y_i)}{\partial W_{r,c}} = 0$$
$$\sum_r \frac{\partial l(x_i, y_i)}{\partial b_r} = 0. \tag{13}$$

That is, the gradients of the bias in the last layer must sum to zero, and the sum of each column in the weight gradients of the last layer must also be zero. Since the sum of gradients for each data point is zero, averaging across all data points in a batch will also yield zero:

$$\sum_r \overline{\nabla b_r} = \frac{1}{B} \sum_{i=1}^{B} \sum_r \frac{\partial l(x_i, y_i)}{\partial b_r} = 0$$
$$\sum_r \overline{\nabla W_{r,c}} = \frac{1}{B} \sum_{i=1}^{B} \sum_r \frac{\partial l(x_i, y_i)}{\partial W_{r,c}} = 0. \tag{7}$$

Consequently, applying the constraint in Equation 7 column by column allows us to filter out incorrect candidate solutions and obtain the true gradients in the phase after the initial training stage.

## A.2. Proof of conditions in the initial training phase

For any row index r, the average gradients of the bias across a batch of data can be denoted as:

$$\overline{\nabla b_r} = \frac{1}{B} \sum_{i=1}^{B} \nabla b_{i,r} = \frac{1}{B} \sum_{j=1}^{T} |\mathbb{B}_j| \overline{\nabla b_{r,\mathbb{B}_j}}, \tag{14}$$

where $\overline{\nabla b_{r,\mathbb{B}_j}} = \frac{1}{|\mathbb{B}_j|} \sum_{i \in \mathbb{B}_j} \nabla b_{i,r}$, (for any $\mathbb{B}_j \neq \emptyset$) is the averaged gradients across the subset $\mathbb{B}_j$. According to Equation 12, we have

$$\overline{\nabla b_{r,\mathbb{B}_j}} = \begin{cases} \frac{1}{|\mathbb{B}_j|} \sum_{i \in \mathbb{B}_j} p_{i,r} - 1 & \text{if } j = r \\ \frac{1}{|\mathbb{B}_j|} \sum_{i \in \mathbb{B}_j} p_{i,r}, & \text{if } j \neq r \end{cases}. \tag{15}$$

Since $p_{i,r} >= 0$, we obtain

$$|B|\overline{\nabla b_r} >= -|\mathbb{B}_r|. \tag{16}$$

For any row index $r$, the average gradients of the weights across a batch of data can be denoted as:

$$\overline{\nabla W_r} = \frac{1}{B} \sum_{i=1}^{B} \nabla W_{i,r} = \frac{1}{B} \sum_{j=1}^{T} |\mathbb{B}_j| \overline{\nabla W_{r,\mathbb{B}_j}}, \tag{17}$$

where $\overline{\nabla W_{r,\mathbb{B}_j}}$, is the averaged gradients across the subset $\mathbb{B}_j$. Combining Equation 12, we have

$$\overline{\nabla W_{r,\mathbb{B}_j}} = \begin{cases} \frac{1}{|\mathbb{B}_j|} \sum_{i \in \mathbb{B}_j} (p_{i,r} - 1) z_i^T & \text{if } j = r \\ \frac{1}{|\mathbb{B}_j|} \sum_{i \in \mathbb{B}_j} p_{i,r} z_i^T, & \text{if } j \neq r \end{cases}. \tag{18}$$

Based on Assumption 3.1, each element in $z_i$ is not negative. Additionally, $p_{i,r} >= 0$. Consequently, for any row index $r$, if $|\mathbb{B}_r| = 0$, $\overline{\nabla W_r} >= 0$. Therefore, we obtain that, for any row index $r$ and column index $c$,

$$\overline{\nabla W_{r,c}} < 0 \Rightarrow \mathbb{B}_r \neq \emptyset. \tag{19}$$

On the basis of Approximation 3.1, we can further simplify the Equation 14 to be:

$$\overline{\nabla b_r} = \frac{1}{B} \sum_{i=1}^{B} \nabla b_{i,r} \approx \frac{|\mathbb{B}_r|}{B} \overline{\nabla b_{r,\mathbb{B}_r}}. \tag{20}$$

Since $p_{i,r} \in [0,1]$, and in the initial training stage, $p_{i,r}$ is typically much smaller than 1, $\frac{1}{|\mathbb{B}_j|} \sum_{i \in \mathbb{B}_j} p_{i,r} - 1 < 0$. Combining Equation 20 and Equation 15, we obtain

$$\overline{\nabla b_r} < 0 \iff \mathbb{B}_r \neq \emptyset. \tag{21}$$

Additionally, as in the initial training stage, $p_{i,r} - 1 \approx -1$, for $|\mathbb{B}_r| \neq 0$, we have

$$\overline{\nabla b_r} \approx \frac{|\mathbb{B}_r|}{B} \overline{\nabla b_{r,\mathbb{B}_r}} = \frac{1}{B} \left( \sum_{i \in \mathbb{B}_r} p_{i,r} - |\mathbb{B}_r| \right) \approx -\frac{|\mathbb{B}_r|}{B}. \tag{22}$$

Finally, combining Equations 7, 16, 19, 21, and 22, we obtain

$$\begin{cases} \sum_r \overline{\nabla b_r} = 0, \\ \sum_r \overline{\nabla W_{r,c}} = 0, \\ |B|\overline{\nabla b_r} >= -|\mathbb{B}_r| \\ \overline{\nabla W_{r,c}} < 0 \Rightarrow \mathbb{B}_r \neq \emptyset \\ (|B|\overline{\nabla b_r} \approx -|\mathbb{B}_r|) \wedge (\overline{\nabla b_r} < 0) \iff \mathbb{B}_r \neq \emptyset. \end{cases}$$

## A.3. Solution of Equation 5

Given the equation for the final-layer gradients formulated in Section 3.2 [1]:

$$\mathcal{U}(\nabla\theta, s) = \frac{\frac{\beta_1}{1-\beta_1^t} m_{t-1} + \frac{1-\beta_1}{1-\beta_1^t} \nabla\theta}{\sqrt{\frac{\beta_2}{1-\beta_2^t} v_{t-1} + \frac{1-\beta_2}{1-\beta_2^t} \nabla\theta^2} + \epsilon}, \tag{5}$$

we let

$$\alpha' = \frac{\beta_1}{1-\beta_1^t} m_{t-1}, \quad \alpha = \frac{1-\beta_1}{1-\beta_1^t},$$

$$\gamma' = \frac{\beta_2}{1-\beta_2^t} v_{t-1}, \quad \gamma = \frac{1-\beta_2}{1-\beta_2^t}, \quad U = \mathcal{U}(\nabla\theta, s).$$

---

[1]In this subsection, $\nabla\theta$ denotes an element of the gradients, specifically $\nabla\theta_w$. For simplicity, we omit the subscript in our notation.

Then Equation 5 (with $\epsilon \approx 0$ neglected) can be rewritten as

$$U = \frac{\alpha' + \alpha \, \nabla\theta}{\sqrt{\gamma' + \gamma \, (\nabla\theta)^2}}. \tag{23}$$

Rearranging and squaring both sides yields a quadratic equation:

$$(U^2 \, \gamma - \alpha^2) \, \nabla\theta^2 \; - \; 2 \, \alpha' \, \alpha \, \nabla\theta \; + \; \left(U^2 \, \gamma' - (\alpha')^2\right) \; = \; 0.$$

Solving for $\nabla\theta$ gives

$$\nabla\theta \; = \; \frac{2 \, \alpha' \, \alpha \; \pm \; \Phi}{2 \, \left(U^2 \, \gamma - \alpha^2\right)}, \tag{24}$$

where

$$\Phi \; = \; \sqrt{\left(2 \, \alpha' \, \alpha\right)^2 \; - \; 4 \, \left(U^2 \, \gamma - \alpha^2\right) \left(U^2 \, \gamma' - (\alpha')^2\right)}.$$

Since $\sqrt{\gamma' + \gamma(\nabla\theta)^2}$ is nonnegative, we choose the root ensuring $\alpha' + \alpha \, \nabla\theta$ shares the same sign as $U$.

## B. Additional Implementation Details

For all attack experiments, we set the batch size to $B = 4$ by default, except in Section 4.3, where different batch sizes are evaluated, and Section 4.4, where $B$ is set to 1 to maximize the effect of previous attacks. Since duplicate labels increase the difficulty of reconstruction (Ye et al., 2024), we ensure this setting in our Section 4 experiments. When $B \geq 4$, each batch contains labels corresponding to at least two distinct data points. Except for the experiments in Appendix C and D, where the impact of the training stage is explored, FL model parameters are randomly initialized following previous works (Zhu et al., 2019; Zhao et al., 2020; Fang et al., 2023; Geiping et al., 2020; Jeon et al., 2021), and the optimizer state after the first step of the second training epoch is used to simulate the target client's optimizer state. We use Adam for image optimization, initializing the learning rate at 0.1 and decaying it at $\frac{3}{8}$, $\frac{5}{8}$ and $\frac{7}{8}$ of the total training iterations. The generated images are initialized following a Gaussian distribution and optimized based on the sign of their gradients computed using Equation 3. All experiments were conducted using NVIDIA RTX A5000, TITAN RTX, and GeForce RTX 4090 GPUs.

## C. Effect of Training Stage on Image Reconstructions

For the experiments in our main paper, we used a randomly initialized model. In this section, we evaluate how different training stages affect the image reconstructions of our attack. Specifically, we train the model on ImageNet (Deng et al., 2009) from scratch and record the model parameters and optimizer state after the $1^{st}, 10^{th}, 100^{th}, 1000^{th}, 10,000^{th}$, and $100,000^{th}$ training iterations. We then conduct attack experiments using the model parameters and optimizer state from each of these stages. The batch size is set to 4, and the ground truth images are identical to those shown in Figure 4. Figure 9

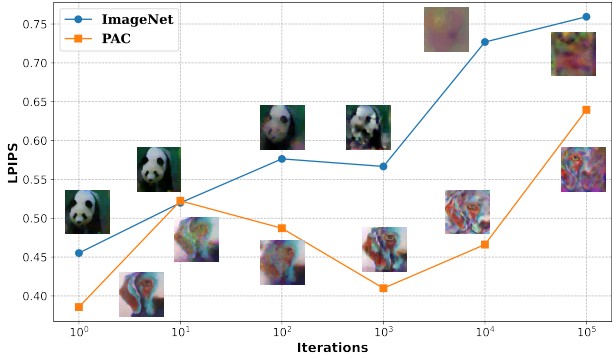

*Figure 9.* The impact of different training stages on LPIPS value.

illustrates the performance of our method at different training stages, evaluated using the average LPIPS values computed with AlexNet. As training progresses, the LPIPS value generally increases, reflecting a gradual decrease in similarity between the reconstructed and real images. Specifically, the reconstructed images exhibit high similarity to the real images

for the model immediately after the first iteration, similar to results obtained with an untrained model. In contrast, after 100,000 training iterations, our method reveals little information about the ground truth.

Besides, for a trained model and its corresponding optimizer state, reconstructions on the PACS dataset are significantly better than those on the ImageNet dataset. For instance, at the $10,000^{th}$ iteration, the generated data for ImageNet conveys virtually no information, whereas the generated data for PACS still retains some similarity to the real data. We attribute this to the PACS dataset having a distribution distinct from that of the ImageNet dataset used during model training. This difference likely results in larger gradient magnitudes for PACS data on the trained model, making it easier to match model updates and reveal information about the real data.

## D. Effect of Training Stage on Candidate Gradient Values

We experimentally investigated the relationship between the number of elements with two candidate solutions after solving Equation 5 and the model training process. Specifically, we trained ResNet18 and ResNet34 (He et al., 2016) models from scratch on the ImageNet dataset using the Adam optimizer. At various training stages, we recorded the model parameters and optimizer states and randomly sampled batches of size 32 from the dataset to compute the corresponding gradients. As shown in Figures 10, the number of elements with only one candidate solution increases rapidly as training progresses. This observation indicates that, beyond the initial training phase, solving Equation 5 alone determines the values of most, but not all, gradient elements. The division of label inference into two cases, as discussed in our main paper, is motivated by this finding.

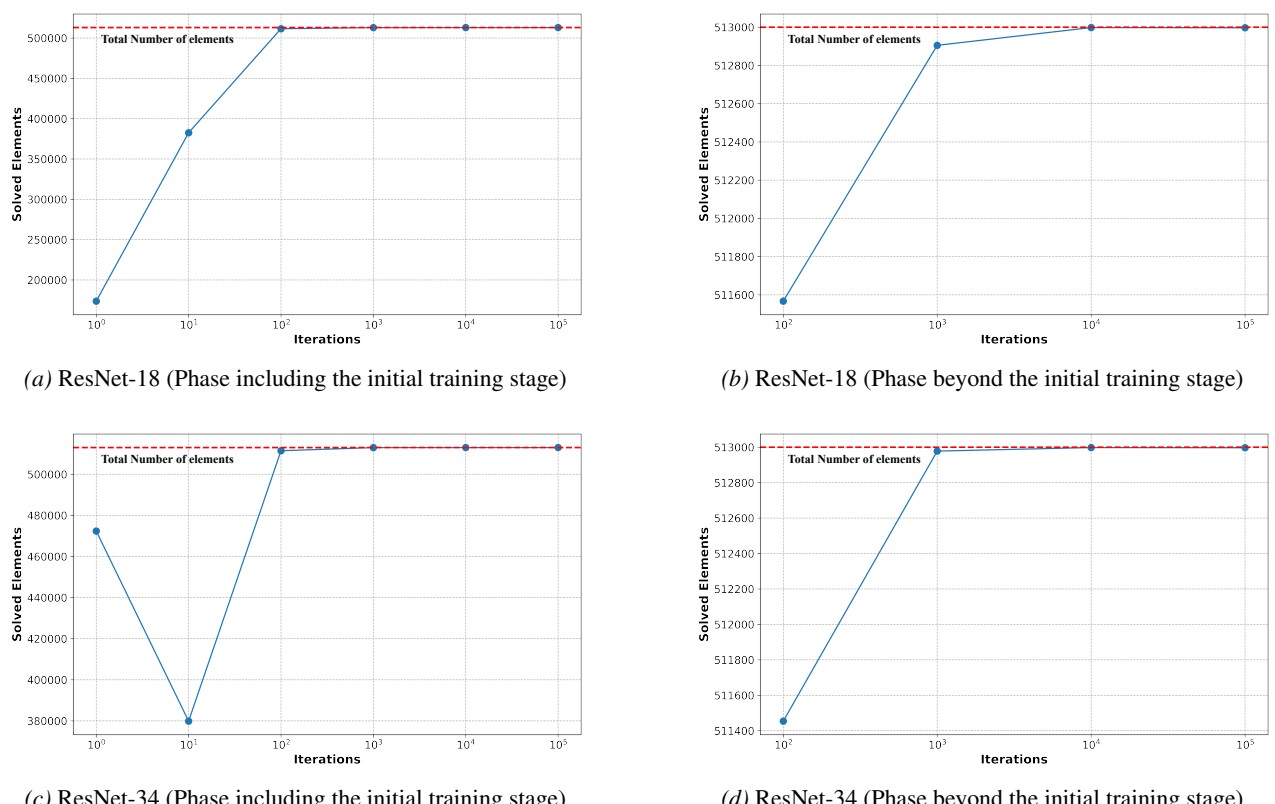

*(a)* ResNet-18 (Phase including the initial training stage)

*(b)* ResNet-18 (Phase beyond the initial training stage)

*(c)* ResNet-34 (Phase including the initial training stage)

*(d)* ResNet-34 (Phase beyond the initial training stage)

*Figure 10.* Number of elements in final layer gradients can be uniquely determined by solving equation 5.

Notably, after the initial training stage, we experimentally observe that the remaining elements with two candidate solutions tend to cluster within specific rows, often corresponding to certain input data labels. The gradient values in these rows can be crucial for estimating the number of data samples associated with the corresponding labels. Therefore, while most elements can be determined solely by solving Equation 5, it remains necessary to resolve the values of the remaining unsolved elements for label inference.

## E. Label Inference Accuracy

**Activation functions.**  We evaluate the label inference accuracy of our method under different activation functions. We consider ReLU, which satisfies Assumption 3.1, and GeLU and SiLU, which do not directly satisfy the assumption but are lower-bounded. Experiments are conducted on ImageNet with ResNet-18 using the Adam optimizer. for GeLU and Swish, we replace the original ReLU activations in ResNet-18 and use the threshold $\tau = 10^{-3}$. We use the model and optimizer states after 100 training steps, and perform label inference on the ImageNet validation set with batch size 32. A prediction is counted as correct only if all labels in the batch are correctly recovered.

*Table 5.* Label inference accuracy under different activation functions on ImageNet.

| Activation | ReLU | GeLU | SiLU |
|---|---|---|---|
| Accuracy (%) | 99.94 | 99.30 | 78.87 |

As shown in Table 5, the proposed method almost perfectly recovers the labels under ReLU, achieving $99.94\%$ accuracy. For GeLU and Swish, although the activations do not directly satisfy Assumption 3.1, their lower-bounded property allows the thresholded criterion to retain informative label recovery. Consequently, the method still achieves $99.30\%$ accuracy with GeLU and $78.87\%$ with Swish, showing that label inference remains effective beyond the exact assumption.

**Additive noise.**  We also evaluate our label recovery method under additive Gaussian noise. Experiments are conducted on ImageNet with ResNet-18 using the Adam optimizer, where the model and optimizer states are taken after 100 training steps and the batch size is 32. We add Gaussian noise with different standard deviations $\sigma$ to the gradients and apply our method. Assuming the attacker knows the noise standard deviation, as is typical in DP settings (Li et al., 2022b), we set $\tau = 5\sigma$ in $\nabla W_{r,c} < -\tau \Rightarrow B_r \neq \emptyset$.

*Table 6.* Label inference accuracy under additive Gaussian noise.

| Noise std ($\sigma$) | 0 | 0.001 | 0.01 | 0.1 |
|---|---|---|---|---|
| Accuracy (%) | 99.94 | 92.32 | 52.94 | 0.00 |

As shown in Table 6, increasing the standard deviation of the Gaussian noise degrades our label inference method, from near-perfect recovery without noise to $0.00\%$ accuracy when $\sigma = 0.1$. However, at $\sigma = 0.01$, our method still achieves $52.94\%$ accuracy, indicating that moderate Gaussian noise remains insufficient to completely prevent label inference by our method.

## F. Experiments with Multiple Local Updates

We further evaluate our method when the client performs multiple local update steps. For label recovery, we approximate the accumulated local updates as a single Adam update by constructing the pseudo-update $\tilde{u} = \Delta\theta/(\eta K)$, where $\Delta\theta$ denotes the total parameter update after $K$ local steps, $\eta$ is the local learning rate, and $K$ is the number of local update steps. This pseudo-update is used for label recovery, while image reconstruction is performed by matching the true accumulated updates.

*Table 7.* Reconstruction performance of our method under multiple local update steps.

| Local Steps | PSNR ↑ | LPIPS ↓ | SSIM ↑ | MSE ↓ |
|---|---|---|---|---|
| 1 | 24.19 | 0.2676 | 0.5940 | 0.0038 |
| 2 | 19.31 | 0.4591 | 0.5064 | 0.0118 |
| 4 | 19.67 | 0.4673 | 0.5158 | 0.0108 |
| 8 | 17.52 | 0.5797 | 0.4718 | 0.0177 |

We find that reconstruction under multiple local updates becomes more effective when the local learning rate is relatively small and when there are no duplicate labels within a batch. We conduct experiments under such a setting with multiple

full-batch local steps, using batch size 2 and learning rate $1 \times 10^{-4}$. All other configurations follow the main experimental setup. The quantitative results are reported in Table 7. As shown in Table 7, while reconstruction quality degrades compared to the single-step case, our method can still reconstruct images that are visually similar to the original images in this multi-step setting: even with up to 8 local steps, the reconstructed images still achieve a PSNR of 17.52 with respect to the ground truth.

## G. Effect of Random Initialization

For the experiments in our main paper, for each reconstruction, we perform four separate runs using different randomly initialized dummy images and select the result with the lowest reconstruction loss as the final output, following prior works (Zhu et al., 2019; Fang et al., 2023; Geiping et al., 2020). Since the effectiveness of gradient inversion attacks can be affected by random initializations, we evaluate our method's robustness to this factor in this section. Specifically, we further conduct five independent runs for the experiments in Section 4.1. For each run, we used a different random seed to initialize both the model parameters and the dummy images. Unlike our main experiments, each run in this setting uses only a single set of initialized dummy images without multiple trials. The evaluation results are summarized in Table 8.

*Table 8.* The impact of random initialization. The *Mean* column reports the average over five runs, with the values in parentheses (*(Main paper result)*) denoting the corresponding results reported in the main paper (results with the lowest reconstruction loss). *SD* and *SE* denote standard deviation and standard error, respectively.

| Metric | Mean (Main paper result) | SD | SE |
|---|---|---|---|
| LPIPS-V ($\downarrow$) | 0.5674 (0.5324) | 0.0316 | 0.0141 |
| LPIPS-A ($\downarrow$) | 0.4818 (0.4230) | 0.0517 | 0.0231 |
| PSNR ($\uparrow$) | 19.8645 (20.5985) | 0.8519 | 0.3810 |
| MSE ($\downarrow$) | 0.0115 (0.0099) | 0.0017 | 0.0008 |
| SSIM ($\uparrow$) | 0.4900 (0.5026) | 0.0168 | 0.0075 |

As shown, compared with the setting in the main paper, where we perform four trials and select the one with the lowest reconstruction loss, the five independent single-trial runs (each with a different checkpoint and randomly initialized dummy image set) show only a slight drop in average performance. Moreover, the variability across random seeds is modest: LPIPS-V $0.5674 \pm 0.0316$, LPIPS-A $0.4818 \pm 0.0517$, PSNR $19.8645 \pm 0.8519$ dB, MSE $0.0115 \pm 0.0017$, and SSIM $0.4900 \pm 0.0168$ (mean $\pm$ SD). This indicates that different seeds yield similar outcomes, and the reconstructions consistently resemble the ground truth closely. These results demonstrate the good repeatability of our method under random initialization.

## H. Runtime Comparison

We report the runtime per reconstruction under the experimental setting described in Section 4, all measured in our experimental environment. Each runtime is measured for a single reconstruction without restart with a single GPU. As shown in Table 9, compared to IG, since our method needs to compute an adaptive optimizer update at each loss computing step, it requires more time (from 0.16h to 0.36h). However, it remains lightweight in absolute terms and is comparable to HFGI. Moreover, compared with approaches that introduce an additional generative model (GIFD and GIAS), our method is more lightweight, requiring less running time.

*Table 9.* Runtime per reconstruction.

| | IG | GIAS | GIFD | HFGI | Ours |
|---|---|---|---|---|---|
| GPU Hour (h) | 0.16 | 0.97 | 0.64 | 0.37 | 0.36 |

## I. Effect of Label Duplication

Prior work has shown that duplicate labels increase the difficulty of gradient-based reconstruction (Ye et al., 2024). We therefore examine how the degree of label duplication within a mini-batch affects our method. In the main paper setting, two out of four images share the same label (pairwise duplication). Here, we vary the duplication level (e.g., all labels distinct,

two pairs, and all labels identical) and report the corresponding reconstruction performance in Table 10. As shown, greater label duplication leads to a slight degradation in reconstruction quality. Nevertheless, even in the extreme case where all four samples share the same label, our method still achieves a reasonably high similarity to the original data. Specifically, the reconstructed images retain a PSNR of 19.80 dB.

*Table 10.* Effect of label duplication on reconstruction quality. The numbers in the 1st column indicate within-batch label multiplicities; e.g., 3+1 means three samples share the same label and one differs.

| Label Duplication | LPIPS-V ↓ | LPIPS-A ↓ | PSNR (dB) ↑ | MSE ↓ | SSIM ↑ |
|---|---|---|---|---|---|
| All-same | 0.6222 | 0.5701 | 19.80 | 0.0111 | 0.4929 |
| 3+1 | 0.5542 | 0.4800 | 20.91 | 0.0090 | 0.5203 |
| 2+2 | 0.5324 | 0.4230 | 20.60 | 0.0099 | 0.5026 |
| All-different | 0.4928 | 0.3949 | 23.90 | 0.0042 | 0.5798 |

## J. More Reconstruction Results

We show additional reconstruction results on images from ImageNet (Deng et al., 2009), PACS (Li et al., 2017), and Web (Ye et al., 2024) in Figure 11. For PACS, the reconstruction results for images in the art painting style are reported in our main paper. In this section, we extend our experiments to include photo and cartoon-style images. We can find that the reconstructed images remain similar to these additional ground truth images, further demonstrating the effectiveness of our methods across various datasets.

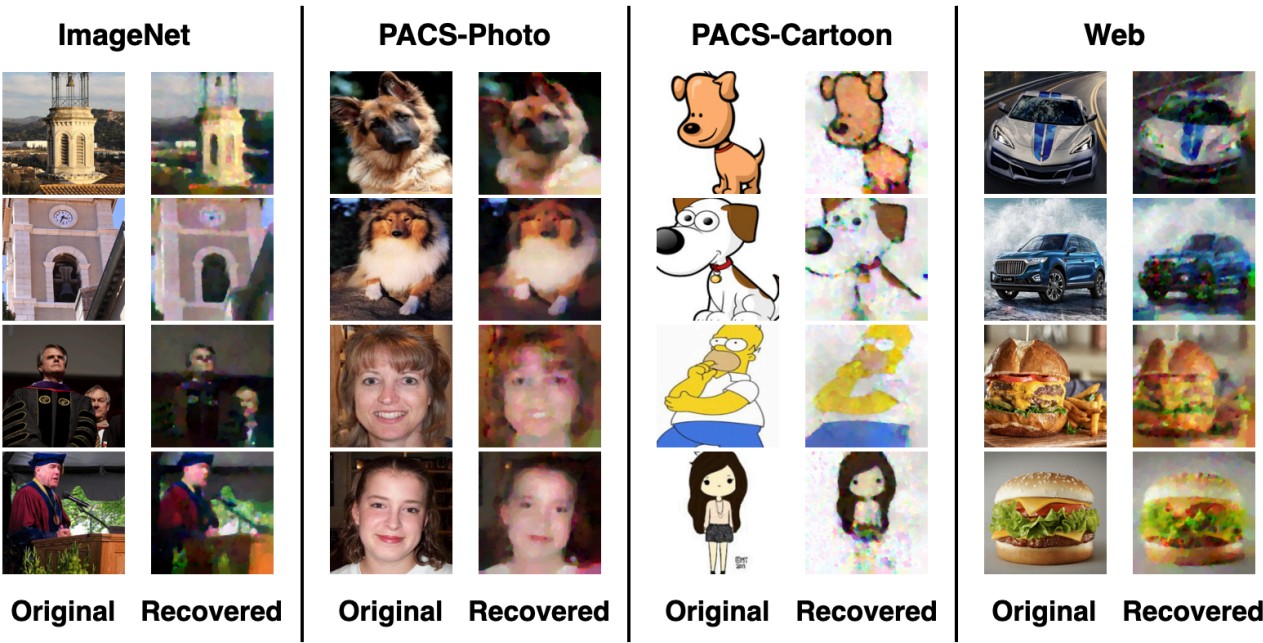

*Figure 11.* More reconstruction results on images from ImageNet, PACS, and Web.

## K. Limitations and Potential Defense

**Limitations.** Following prior works (Zhu et al., 2019; Fang et al., 2023; Geiping et al., 2020), we assume an honest-but-curious server that can legitimately access per-client model updates. Our study does not consider other threat models. For instance, a malicious server that actively manipulates the global model may be able to amplify the privacy-leakage risks, and FL systems employing MPC-based secure aggregation can preclude reliable access to per-client updates (Bonawitz et al., 2016), thereby substantially weakening our attack. We regard a systematic study of these scenarios as a promising direction for future work. In addition, as shown in Figure 6, our reconstruction quality degrades as the local batch size

increases. For sufficiently large batches, our attack becomes ineffective. Moreover, our current analysis and experiments mainly consider the case where the number of local updates is equal to one. We regard a more comprehensive study of the multi-step setting as an important direction for future work. Furthermore, our analytical derivation relies on certain assumptions and approximations. While they are consistent with our empirical observations in the evaluated settings, they may not hold equally well for all architectures, activation functions, training stages, or data distributions. Our method also relies on the assumption that the attacker has access to the initial state of the local optimizer. As a result, the current method does not apply to settings where such states are not shared.

**Potential defenses.**    Because our method relies on access to per-client updates, reliable multi-party computation protocols can substantially diminish its effectiveness. However, such defenses incur substantial computation and communication overhead, introducing a trade-off between utility and safety. Besides, previous studies have also discussed possible avenues to circumvent such protocols (Pasquini et al., 2022). A simpler defense is to use a large local batch size: increasing the local batch size reduces per-sample signal and tends to weaken our attack. For sufficiently large batches, effective reconstruction can no longer be feasible.

