# OpenReview forum: "Gradient Inversion Attacks Beyond SGD"
_ICML.cc/2026/Conference — ICML 2026 regular_

### Official Review · Reviewer_2yEx · 2026-02-20

**Soundness:** 3
**Presentation:** 3
**Significance:** 3
**Originality:** 2
**Overall Recommendation:** 4
**Confidence:** 5

**Summary:**

The paper presents an optimization-based gradient inversion attack in federated learning, designed to recover training data when clients employ a client-adaptive update rule rather than directly sharing gradients. The authors' main contributions include:
- An "update matching" objective function that incorporates the state of the optimizer instead of solely minimizing the gradient distance.
- A two-step method to recover labels from updates, which first extracts the gradients of the final classification layer from updates by solving quadratic equations, and then applies existing techniques to recover the labels.

Experimental results on image datasets show improved reconstruction quality compared to existing gradient matching methods.

**Compliance With Llm Reviewing Policy:**

Affirmed.

**Final Justification:**

The paper presents an optimization-based gradient inversion attack in federated learning, designed to recover training data when clients employ a client-adaptive update rule rather than directly sharing gradients.

Among the strengths, I particularly value the novelty of the approach and the limitation it addresses—namely, extending standard optimization-based attacks to settings with momentum-based updates.

The paper still exhibits some weaknesses, such as a limited evaluation in the presence of defense mechanisms and questions regarding scalability to larger batch sizes, as also acknowledged by the authors. However, since the goal of the work is not to improve the state of the art in these aspects, I believe the contribution remains valuable.

During the rebuttal phase, I had concerns regarding the soundness of the experimental results, the fairness of the baselines, and the novelty of the approach. The authors’ responses helped clarify these aspects and better position both their evaluation and contributions. Therefore, despite some remaining limitations, I believe the work offers a meaningful contribution that can support future research on gradient inversion attacks.

**Key Questions For Authors:**

According to the described weaknesses, my questions are the following:

1. Regarding the experimental evaluation, could the authors clarify the rationale behind the selection of the four baselines?

2. In Table 1, HFGI appears to perform worse than IG. Since HFGI is a more recent method and an improvement over IG, could you explain the reasons behind this performance drop?

3. Could the authors clarify whether HFGI has access to the ground-truth labels in the experiments presented in Section 4.4?

4. Could the authors share their intuition on why Adam appears to yield the best results for reconstruction quality (Tables 1 and  2)? Building on the previous point, would it be possible to include HFGI results in the momentum setting? Seeing those results, especially where gradients could potentially be used for label recovery, would help in having a better overview of the attack performances.

5. Do the authors have any intuition about the behavior of the double-solution issue for Equation 5? In particular, why does the problem appear more challenging during the initial rounds?

6. Following the observation that after a certain number of rounds the double solutions seem to disappear (Figure 10 in the appendix), do the authors think it might be possible in principle to extend Equation 5 to recover gradients beyond the last layer, potentially enabling recovery of the full gradient?

**Limitations:**

Although the authors address limitations such as the need for model update access and performance degradation with larger batch sizes, the discussion remains somewhat incomplete. The paper would be strengthened by explicitly discussing how the attack's feasibility and accuracy scale with an increased number of local epochs and multiple minibatch updates. Additionally, a brief discussion on the attack's robustness against differential privacy would provide a more complete picture of its practical threat.

**Strengths And Weaknesses:**

### Soundness

*Strengths*

The mathematical analysis of the paper is technically sound, and the method appears correct.

*Weaknesses*

I have a few doubts and suggestions regarding the evaluation section, particularly the experimental setup and the interpretation of the results. I believe clarifying these points would help strengthen the empirical validation of the paper.

1. While the selected baselines provide a useful point of comparison, I would appreciate clarification on the rationale behind their selection. In particular, the evaluation includes an old optimization-based attack, a more recent one, and two generative approaches. It would have been helpful to compare against more recent optimization-based attacks (e.g., [1], [2], [3], [4]), or to discuss how these methods relate to the proposed approach.


2. I understand the choice to not provide labels to competing attacks, as label recovery is part of the contribution and this design choice seems well motivated. However, I found it somewhat unexpected that HFGI performs worse than IG in Table 1, since HFGI is a later improvement of IG.

3. I am also interested in the results obtained with different optimizers (Table 2). In particular, the performance with momentum appears to be worse than with Adam for both attacks, which intuitively, I did not expect. For example, in the case of IG, momentum should in principle make it possible to extract the gradients and match the original loss, leading to better results. Additionally, including HFGI results in this table—especially with momentum, where gradients could potentially be used for label recovery—could provide further insight.


4. Finally, it is not entirely clear to me whether HFGI has access to the labels in the experiments in Section 4.4. Since the attacker has access to gradients, I guess that label recovery could be applied, and clarifying this mechanism explicitly would make the experimental results easier to interpret.

[1] Yin, H., Mallya, A., Vahdat, A., Alvarez, J. M., Kautz, J., and Molchanov, P. See through gradients: Image batch recovery via gradinversion

[2] Kariyappa, S., Guo, C., Maeng, K., Xiong, W., Suh, G. E., Qureshi, M. K., and Lee, H.-H. S. Cocktail party attack: Breaking aggregation-based privacy in federated learning using independent component analysis.

[3] Li, Z., Wang, L., Chen, G., Zhang, Z., Shafiq, M., and Gu, Z. E2EGI: End-to-end gradient inversion in federated learning

[4] Li, B., Gu, H., Chen, R., Li, J., Wu, C., Ruan, N., Si, X., and Fan, L. Temporal gradient inversion attacks with robust optimization



### Presentation

*Strengths*

The paper is generally well written, and the goals of the work are clearly stated. The paper is well positioned with respect to the existing literature and addresses a clearly identified gap.

*Weaknesses*

I have only a few minor presentation comments. For example, when introducing Equation 7, it might be helpful to include an explicit reference to the corresponding proof in the appendix at that point, rather than only mentioning it at the end of the paragraph. This would make the structure easier to follow. Additionally, in Section 3.2, clarity could be improved by using distinct notation to differentiate between gradients of the full model and gradients of the last layer, as the current notation can be somewhat confusing. Also, when referring to the appendix in this section, it would be helpful to point to the specific figure (Figure 10) or subsection.

Finally, whenever possible, ensure to reference the published version of papers rather than the arXiv version. Examples include, but are not limited to, Fowl et al. (2021) and Du et al. (2024).

### Significance

*Strengths*

The paper addresses a current limitation of existing optimization-based attacks by considering the setting of client-adaptive optimizers in Federated Learning. The results suggest that these optimizers may also be vulnerable to gradient inversion attacks, which helps broaden the understanding of privacy risks in this setting and may encourage further work on this specific class of reconstruction attacks.

*Weaknesses*

That said, the applicability of the attack appears very limited at this stage. In particular, the evaluation focuses on scenarios with a single local update and relatively small batch sizes, and considers only the specific class of client-adaptive optimizers.

### Originality

*Strengths*

The paper introduces a label reconstruction approach that improves reconstruction quality in the considered setting, which is an interesting contribution.

*Weaknesses*

At the same time, the method builds substantially on prior work. The optimization objective in Equation 3 appears to be a natural extension of the formulation proposed in IG (Geiping et al., 2020), and the label reconstruction part applies existing techniques (Ma et al., 2023; Ye et al., 2024). As a result, the main novelty seems to lie in the formulation and solution of Equation 5.

---

> ### Author Rebuttal · Authors · 2026-03-31
>
> We truly appreciate the reviewer's constructive comments.
>
> > W1 & Q1: Rationale for baseline selection.
>
> IG is one of the first GIAs to achieve high-resolution (e.g., ImageNet-scale) reconstruction. GIAS and GIFD represent an important line of work that improves GIA using generative priors. HFGI is a more recent approach that achieves high-fidelity reconstruction from raw gradients without relying on generative models. Together, these baselines include both classical and recent optimization-based methods, with and without generative priors.
>
> We thank the reviewer for pointing out the additional relevant works and will include them in the revision. [1] improves reconstruction using BN statistics, [2] recovers inputs from aggregated gradients, [3] enhances inversion via analytic reconstruction and sample matching, and [4] leverages temporal gradients to improve attacks. However, these methods are still designed for the standard SGD setting where raw gradients are accessible.
>
> > W2 & Q2: Comparison between IG and HFGI.
>
> Although HFGI is a more recent method, its improvements are mainly designed for the standard SGD setting, thus showing no advantage over IG under adaptive optimizers. Under SGD (Table 3 and Fig. 7), HFGI outperforms IG, consistent with its original design.
>
> > W4 & Q3: Whether HFGI has access to ground-truth labels in Section 4.4.
>
> Yes. In Section 4.4, HFGI can recover the correct labels from accessible gradients.
>
> > W3 & Q4 (1): HFGI results in the momentum setting.
>
> We report the performance of HFGI and our method under both momentum and Adam in the Table below. Under momentum, HFGI achieves reconstruction quality comparable to ours. Moreover, HFGI performs substantially better with momentum than with Adam, while our method performs better with Adam.
>
> | Optimizer | Method | PSNR ↑ | MSE ↓ | SSIM ↑ | LPIPS ↓ |
> |----------|--------|--------|--------|--------|---------------|
> | Momentum | HFGI (gradient accessible) | 15.88 | 0.03 | 0.41 | 0.57 |
> | Momentum | Ours   | 17.01 | 0.02 | 0.41 | 0.56 |
> | Adam     | HFGI   | 7.01  | 0.20 | 0.23 | 1.21 |
> | Adam     | Ours   | 20.60 | 0.01 | 0.50 | 0.42 |
>
> > Q4 (2): Intuition on why Adam appears to perform best (Tables 1 and 2).
>
> Intuitively, Adam produces more structured and normalized updates via moment estimates, which reduces scale variability and stabilizes the optimization landscape. This makes the update signal more amenable to matching, resulting in improved reconstruction quality.
>
> > The optimization objective in Equation 3 appears to be a natural extension of the formulation proposed in IG. The main novelty seems to lie in the formulation and solution of Equation 5.
>
> While Eq. 3 can be viewed as an extension of prior gradient matching formulations, its effect goes beyond a straightforward extension. In particular, under the standard SGD setting (Section 4.4), our method does not reduce to prior approaches and instead achieves stronger reconstruction performance, with a 3.0 dB PSNR improvement over baselines (Table 3 and Fig. 7). This improvement directly stems from the proposed objective.
>
> In addition, we clarify that the novelty is not limited to Eq. 5, but lies in enabling gradient inversion from updates of adaptive optimizers, which requires jointly addressing update matching and label reconstruction under this new setting. The proposed components work together to make reconstruction feasible when raw gradients are not accessible.
>
> > Q5: Intuition on the double-solution issue in Eq. 5.
>
> Intuitively, in early rounds, Adam updates are close to a sign-normalized transform of the gradient, carrying limited magnitude information, which makes the two roots less distinguishable. Thus, the spurious root is harder to filter out. As training progresses, accumulated moment estimates restore magnitude sensitivity, making the correct root easier to identify.
>
> > Q6: Can Eq. 5 be extended beyond the last layer to recover full gradients?
>
> While the number of double solutions decreases rapidly after a certain number of rounds, it may not vanish completely. Thus, Eq. 5 can be extended beyond the last layer to recover gradients at other layers, but does not reliably enable full gradient recovery on its own.
>
> > Discussion on the attack's robustness against differential privacy (DP).
>
> We evaluate the robustness of our label inference under additive Gaussian noise, a standard mechanism used in DP. Experiments are conducted on ResNet-18 with ImageNet using Adam, with model states taken after 100 training steps and batch size 32. The results are shown in the Table below. As shown, the accuracy of our label inference degrades as noise increases, but still exceeds 50% at $\sigma = 0.01$.
>
> | Noise std ($\sigma$) | 0 | 0.001 | 0.01 | 0.1 |
> |---------------------|----|--------|-------|------|
> | Label inference accuracy (%) | 99.94 | 92.32 | 52.94 | 0.00 |
>
> > Presentation suggestion.
>
> We thank the reviewer for these suggestions and will incorporate them in the revision.

---

> > ### Author Rebuttal · Reviewer_2yEx · 2026-04-03
> >
> > I thank the authors for their responses and for providing additional experiments. I appreciated their clarification regarding the novelty of the work, which helped me better position and understand the value of their contribution. That said, my concerns have now been fully addressed, and I will raise my score to 4.

---

> > > ### Author Response · Authors · 2026-04-04
> > >
> > > We truly appreciate the reviewer for the valuable comments and for agreeing to raise the score. We would like to kindly remind the reviewer about the score update when convenient, as mentioned in the last comments. Thanks!

---

### Official Review · Reviewer_knSg · 2026-03-03

**Soundness:** 3
**Presentation:** 4
**Significance:** 4
**Originality:** 4
**Overall Recommendation:** 4
**Confidence:** 2

**Summary:**

The paper introduces a Gradient Inversion Attack (GIA) method tailored to federated learning (FL) systems that utilize adaptive optimizers (such as Adam, RMSProp, and AdaGrad). Previous GIA research focuses primarily on the standard SGD setting, relying heavily on the assumption that raw gradients are accessible to the attacker. To address the absence of raw gradients in modern FL setups, the authors propose a method that first analytically recovers labels from model updates and then iteratively optimizes generated images to match the true model updates. The method is extensively evaluated on ImageNet and PACS datasets, demonstrating that it significantly outperforms existing baselines (IG, GIAS, GIFD, and HFGI). Furthermore, the authors show that their proposed update-matching objective improves attack performance even in the traditional SGD setting.

**Compliance With Llm Reviewing Policy:**

Affirmed.

**Final Justification:**

The paper tackles an important underexplored threat (adaptive optimizers) with sound methodology and strong empirical gains (+7 dB PSNR). The rebuttal partially addressed my concerns on non‑negative activations and DP robustness. However, key limitations remain: attack effectiveness degrades sharply after ~100k iterations and for batch sizes >16. Given these unresolved weaknesses, the contribution is solid but incremental. Therefore, I keep my original score unchanged.

**Key Questions For Authors:**

See Weaknesses mentioned above.

**Limitations:**

yes

**Strengths And Weaknesses:**

**Strengths:**

1. The authors successfully extend GIAs beyond the standard SGD setting to adaptive optimizers, addressing a highly practical, realistic, and underexplored threat model in modern FL.

2. The proposed method reconstructs images that closely resemble the originals in the adaptive-optimizer setting, a scenario where baseline methods produce barely recognizable outputs. It improves the PSNR by up to 7 dB over the baselines.

3. The approach generalises effectively across various modern adaptive optimizers, including Adam, AdaGrad, and RMSProp.

4. The paper presents a thoughtful analytical method for label recovery by dividing the training phase into initial and post-initial stages and mathematically deriving strict constraints.

**Weaknesses:**

1. How sensitive is the label recovery to defense mechanisms like Differential Privacy, which inject noise into the updates?

2. The initial training phase's label recovery heavily relies on Hypothesis 3.1, which stipulates that the input to the final fully connected layer must be non-negative, implicitly assuming activation functions such as ReLU or Sigmoid. Furthermore, these constraints depend on the presence of a bias term in the last layer. How can this derivation be generalized to modern architectures using activation functions that allow negative values (e.g., GeLU, Swish), or models that omit the bias term in their classification heads?

3. Experimental results show that the effectiveness of the attack is highly dependent on the model's training phase; after 100,000 iterations, the reconstruction results reveal no information. Similarly, as the batch size increases, the reconstruction quality significantly degrades, and when the batch size is greater than 16, the reconstruction almost completely fails.

---

> ### Author Rebuttal · Authors · 2026-03-31
>
> We truly appreciate the reviewer's constructive comments.
>
> > W2: The initial training phase's label recovery heavily relies on Hypothesis 3.1, which stipulates that the input to the final fully connected layer must be non-negative, implicitly assuming activation functions such as ReLU or Sigmoid. How can this derivation be generalized to modern architectures using activation functions that allow negative values (e.g., GeLU, Swish)?
>
> Thanks for the comment. While Assumption 3.1 does not hold for activation functions that allow negative values, the derivation can be relaxed rather than invalidated in this setting.
>
> Specifically, when the activation value $z_{n, i}$ can take negative values, the implication in Eq. (19) ($\overline{\nabla W_{r,c}} < 0 \Rightarrow B_r \neq \emptyset$) does not strictly hold. Nevertheless, the row-wise gradient expression in Eq. (18) remains valid. What changes is the sign-based implication derived from it. We therefore replace the strict sign condition in Eq. (19) with a thresholded criterion:
> $$
> \overline{\nabla W_{r,c}} < -\tau \Rightarrow B_r \neq \emptyset,
> $$
> where $\tau$ is a small constant (we use $10^{-3}$ in our experiments), and $B_r$ denotes the set of samples in the batch belonging to class $r$. This is based on the observation that negative values may appear when class $r$ is absent, but such values are typically much smaller in magnitude compared to the case where $B_r \neq \emptyset$, due to lower-bounded activations and small predicted probabilities in early training.
>
> To evaluate the effectiveness of our generalized method, we conduct experiments with different activation functions. Specifically, we replace the ReLU in ResNet-18 with GeLU and Swish, and perform experiments on ImageNet using the Adam optimizer. We use the model and optimizer states after 100 training steps, and perform label inference on the ImageNet validation set with batch size 32. A prediction is counted as correct only if all labels in the batch are correctly recovered. The results are shown in the table below. Empirically, while the performance does degrade under signed activations, it remains above 78% accuracy in this setting.
>
> | Activation | ReLU | GeLU | Swish |
> |------------|------|------|-------|
> | Accuracy (%)   | 99.94 | 99.30 | 78.87 |
>
>
> > W1: How sensitive is the label recovery to defense mechanisms like Differential Privacy, which inject noise into the updates?
>
> Thanks for the comment. We evaluate the sensitivity of label recovery under additive Gaussian noise. Using the same experimental setup as above, we add Gaussian noise to the gradients and apply our generalized method. Assuming the attacker knows the noise standard deviation, as is typical in DP settings, we set $\tau = 5\sigma$ in $\overline{\nabla W_{r,c}} < -\tau \Rightarrow B_r \neq \emptyset$. The results are shown in the table below. The effectiveness of our method for label inference degrades as the noise level increases, but still exceeds 50% accuracy at $\sigma = 0.01$.
>
> | Noise std ($\sigma$) | 0 | 0.001 | 0.01 | 0.1 |
> |---------------------|----|--------|-------|------|
> | Accuracy (%)            | 99.94 | 92.32 | 52.94 | 0.00 |
>
>
> > W3: Experimental results show that the effectiveness of the attack is highly dependent on the model's training phase. Similarly, as the batch size increases, the reconstruction quality significantly degrades.
>
> Thanks for the comment. We view improving robustness across different training stages and batch sizes as an important direction for future work.

---

> > ### Author Rebuttal · Reviewer_knSg · 2026-04-03
> >
> > I thank the author for the additional experiments, therefore I remains the score unchanged.

---

> > > ### Author Response · Authors · 2026-04-03
> > >
> > > Thank you for your valuable comments. We sincerely appreciate your time and effort in reviewing our submission.

---

### Official Review · Reviewer_9uGa · 2026-03-13

**Soundness:** 3
**Presentation:** 3
**Significance:** 3
**Originality:** 3
**Overall Recommendation:** 4
**Confidence:** 3

**Summary:**

This paper introduces a novel Gradient Inversion Attack targeting FL systems that rely on adaptive optimizers (like Adam). Because these systems share parameter updates instead of raw gradients, traditional gradient-based attacks fall short. To bridge this gap, the authors designed a two-step attack framework: first, it analytically derives the ground-truth data labels directly from the model updates; second, it introduces a novel "update-matching" objective to iteratively optimize dummy images until their resulting updates match the victim's observed signals. Experiments demonstrate that this approach not only successfully reconstructs high-fidelity private images without access to raw gradients, but it also outperforms existing baseline attacks even in standard SGD setups.

**Compliance With Llm Reviewing Policy:**

Affirmed.

**Key Questions For Authors:**

1. The framework assumes the attacker has access to the initial optimizer state, such as momentum . If the FL protocol does not explicitly share these states, how robust is your attack if the server must estimate them purely from historical updates?

2. Assumption 3.1 relies on non-negative activation functions like ReLU for label inference . How does the label recovery perform on modern architectures that utilize activations with negative regimes, such as GELU or SiLU?

3. The derivation in Equation 5 ignores the numerical stability term $\epsilon$ ($\epsilon \approx 0$). Does omitting $\epsilon$ cause numerical instability or affect the accuracy of the derived gradients in later training stages when update magnitudes become extremely small?

4. Given the attack's strict reliance on per-client updates, can this update-matching method be adapted to extract meaningful private data from small-cluster aggregated updates in systems employing partial Secure Aggregation?

5. The reconstruction quality degrades as the local batch size increases. What specific factors cause this bottleneck, and are there potential regularizers or GAN priors that could help sustain the attack's effectiveness at larger, more realistic batch sizes (e.g., $B \ge 64$)?

**Limitations:**

Yes.

**Strengths And Weaknesses:**

# Strangths
1. This study introduces a gradient inversion attack tailored for adaptive optimizers in FL, broadening attack scenarios that were previously largely limited to standard SGD.

2. The proposed method demonstrates better image reconstruction quality compared to existing baselines in both adaptive optimizer and traditional SGD settings.

3. The authors construct a theoretical approach to analytically recover training data labels from model updates by deriving mathematical constraints on final-layer gradients.

# Weaknesses
1. The quality of image reconstruction decreases as the client's local training batch size increases.

2. The effectiveness of the attack is limited if the FL system deploys secure aggregation protocols to hide individual per-client updates.

3. The execution of this framework relies on the assumption that the attacker has access to the initial state of the local optimizer.

4. The method requires more computational time than certain baseline attacks because it must simulate the adaptive optimizer's update at each loss computation step.

5. The presence of duplicate labels within a training batch increases reconstruction difficulty and affects the final image quality.

---

> ### Author Rebuttal · Authors · 2026-03-31
>
> We truly appreciate the reviewer for the constructive comments.
>
> > W1 & Q5: The reconstruction quality degrades as the local batch size increases.
>
> Thanks for the comment. Degradation at larger batch sizes arises because multiple samples jointly contribute to the updates, which increases ambiguity and enlarges the solution space. Nevertheless, as shown in Fig. 6, our method consistently achieves better reconstruction than the baseline (IG) under the same batch size. As this limitation is related to high-dimensional optimization in the image space, a promising direction is to optimize in lower-dimensional spaces (e.g., GAN latent space), thereby simplifying the optimization.
>
> > W3 & Q1: The execution of this framework relies on the assumption that the attacker has access to the initial state of the local optimizer.
>
> Thanks for the comment. Our current attack and theoretical derivation rely on access to the initial optimizer state, which is part of the threat model and consistent with adaptive FL protocols that synchronize optimizer states across clients [1]. As a result, the current method does not directly apply to settings where such states are not shared. We consider extending the attack to this setting an important direction for future work.
>
> > Q2: Assumption 3.1 relies on non-negative activation functions. How does the label recovery perform on modern architectures that utilize activations with negative regimes?
>
> While Assumption 3.1 does not hold for activation functions that allow negative values, the derivation can be relaxed rather than invalidated in this setting.
>
> Specifically, when the activation value can take negative values, the implication in Eq. (19) does not strictly hold. We instead use a thresholded criterion:
> $$
> \overline{\nabla W_{r,c}} < -\tau \Rightarrow B_r \neq \emptyset,
> $$
> where $\tau$ is a small constant (we use $10^{-3}$ in our experiments). This is based on the observation that negative values may appear when class $r$ is absent, but such values are typically much smaller in magnitude compared to the case where $B_r \neq \emptyset$, due to lower-bounded activations and small predicted probabilities in early training.
>
> To evaluate the effectiveness of our generalized method, we conduct experiments with different activation functions. Specifically, we replace the ReLU in ResNet-18 with GeLU and SiLU, and perform experiments on ImageNet using the Adam optimizer. We use the model and optimizer states after 100 training steps, and perform label inference on the ImageNet validation set with batch size 32. A prediction is counted as correct only if all labels in the batch are correctly recovered. The results are shown in the Table below. While the performance does degrade under signed activations, it remains above 78% accuracy in this setting.
>
> | Activation | ReLU | GeLU | SiLU |
> |------------|------|------|-------|
> | Accuracy (%)   | 99.94 | 99.30 | 78.87 |
>
> >Q3: The derivation in Equation 5 ignores the numerical stability term $\epsilon$ ($\epsilon \approx 0$).
>
> Thanks for the comment. The numerical stability term $\epsilon$ can be retained in Eq. (5). It does not change the overall derivation. We omit it for simplicity. We empirically validate its effect under the same setting as above, with model and optimizer states taken after 10,000 training steps. The results are shown in the Table below, from which we observe almost no difference. This is because label inference produces discrete outputs, and small perturbations typically do not cross the discretization boundary and thus do not change the result. Nonetheless, we do not rule out potential effects in theory.
>
> | Setting        | Ignore $\epsilon$ | Include $\epsilon$ |
> |----------------|------------------|-------------------|
> | Accuracy (%)   | 64.34            | 64.34             |
>
> > Q4 & W2: Can this update-matching method be adapted to extract meaningful private data from small-cluster aggregated updates in systems employing partial Secure Aggregation?
>
> Thanks for the comment. We evaluate our method under secure aggregation, where updates from four clients are aggregated and only the aggregated update is observed. The results are shown in the Table below, from which we observe that secure aggregation significantly degrades reconstruction quality.
>
> | Defense  | MSE $\downarrow$ | PSNR $\uparrow$ | SSIM $\uparrow$ | LPIPS-A $\downarrow$ | LPIPS-V $\downarrow$ |
> |----------|----------------|-------------------|-------------------|--------------------|--------------------|
> | No Defense | 0.0099 | 20.5985 | 0.5026 | 0.4230 | 0.5324 |
> | Secure Aggregation | 0.1527 | 8.9056 | 0.1895 | 0.8901 | 0.8805 |
>
> This indicates that our method struggles to extract meaningful private data under secure aggregation. We regard a systematic study of such scenarios as a promising direction for future work.
>
> [1] Mime: Mimicking centralized stochastic algorithms in federated learning.

---

> > ### Author Rebuttal · Reviewer_9uGa · 2026-04-03
> >
> > Overall satisfied, but my main concern aligns with other reviewers (which the authors acknowledged in their rebuttal). Specifically, since current defenses are already very strong, the assumptions this method relies on make it impractical. Nevertheless, it has value as a purely theoretical exploration. Thus, despite my concerns, I rated it as a weak accept.

---

> > > ### Author Response · Authors · 2026-04-04
> > >
> > > Thank you for your valuable comments and for your overall positive assessment. We sincerely appreciate your time and effort in reviewing our submission.

---

### Official Review · Reviewer_23ff · 2026-03-13

**Soundness:** 3
**Presentation:** 4
**Significance:** 2
**Originality:** 4
**Overall Recommendation:** 4
**Confidence:** 4

**Summary:**

The paper shows how to reconstruct private data from model updates computed by the Adam optimizer. The experimental results demonstrate the effectiveness of the proposed method. A theoretical derivation is also provided.

**Compliance With Llm Reviewing Policy:**

Affirmed.

**Final Justification:**

After the rebuttal, both the initial and subsequent concerns have been adequately addressed, leading me to change my recommendation from a weak rejection to full acceptance. Therefore, I revise my score to 4 and recommend accepting the paper.

**Key Questions For Authors:**

1. Why are more recent works (e.g., [1,2,3,4,5]) not mentioned?
2. In around lines 238-240, it says that attackers can obtain $m_{t-1}$ and $v_{t-1}$, while in the Threat Model, the attackers can only access the initial values of $m_{t-1}$ and $v_{t-1}$. So, is the derivation of label recovery based on flawed assumptions?


[1] Shan, Junjie, et al. "Geminio: Language-guided gradient inversion attacks in federated learning." Proceedings of the IEEE/CVF International Conference on Computer Vision. 2025.
[2] Yu, Wenbo, et al. "Gi-nas: Boosting gradient inversion attacks through adaptive neural architecture search." IEEE Transactions on Information Forensics and Security (2025).
[3] Li, Bowen, et al. "Temporal gradient inversion attacks with robust optimization." IEEE Transactions on Dependable and Secure Computing 22.4 (2025): 3383-3397.
[4] Xuan Liu, Siqi Cai, Qihua Zhou, Song Guo, Ruibin Li, and Kaiwei Lin. Mjölnir: Breaking the shield of perturbation-protected gradients via adaptive diffusion.Proceedings of the AAAI Conference on Artificial Intelligence, 39(25):26308–26316, Apr. 2025.
[5] Shanghao Shi, Ning Wang, Yang Xiao, Chaoyu Zhang, Yi Shi, Y. Thomas Hou, and Wen Jing. Lou. Scalemia: A scalable model inversion attack against secure federated learning via latent space reconstruction. In 32nd Annual Network and Distributed System Security Symposium, NDSS, 2025.

**Limitations:**

yes

**Strengths And Weaknesses:**

Strengths
- The paper studies GIAs, which have been well-studied in prior work, but from a perspective that remains unexplored.
- Clear explanations and derivations of the proposed method are provided.
- The proposed method significantly outperforms other baselines.

Weaknesses
- Most of the content is devoted to the Adam optimizer. Discussion of other adaptive optimizers is absent.
- Although this topic may not be studied before, which makes this paper earns some originality, the discussion of the proposed method features the adam optimizer only, which makes this paper's significance marginal.

---

> ### Author Rebuttal · Authors · 2026-03-31
>
> We truly appreciate the reviewer's constructive comments.
>
> > W1 & W2: Most of the content is devoted to the Adam optimizer. The discussion of the proposed method features the Adam optimizer only, which makes this paper's significance marginal.
>
> Thanks for the comment. While we present the method using Adam for ease of exposition, the proposed method is not restricted to a specific optimizer.
>
> Our method is based on update matching and applies to optimizers with element-wise update rules and accessible initial states. Adam is used as a representative example due to its practical relevance and the involvement of both first- and second-order moments. The derivation in Sec. 3 is not specific to Adam. It can be applied in the same manner once the corresponding update rule is substituted.
>
> We also include results on other adaptive optimizers in Sec. 4.2 (Table 2 and Fig. 5), where similar improvements over the baseline are observed. The quantitative results are summarized below.
>
> | Optimizer | Method | PSNR ↑ | LPIPS ↓ | SSIM ↑ | MSE ↓ |
> |-----------|--------|--------|--------|--------|--------|
> | AdaGrad   | IG     | 9.64   | 0.78   | 0.28   | 0.11   |
> |           | Ours   | 18.72  | 0.48   | 0.43   | 0.02   |
> | RMSProp   | IG     | 11.85  | 0.77   | 0.30   | 0.07   |
> |           | Ours   | 20.46  | 0.46   | 0.50   | 0.01   |
>
> > Q1: Why are more recent works (e.g., [1,2,3,4,5]) not mentioned?
>
> We thank the reviewer for pointing out these related works and will include them in the revision.
>
> These works [1,2,3,4,5] are still mainly studied under the SGD-based setting, where gradient information is available to the attacker. In contrast, our work considers the case where clients adopt adaptive optimizers and only the resulting model updates are observable, without direct access to gradients.
>
> > Q2: The derivation uses $m_{t-1}$ and $v_{t-1}$, while the threat model states that the attacker can only access the initial optimizer state. Is the derivation of label recovery based on flawed assumptions?
>
> Thanks for the comments. There is no inconsistency here.
>
> In our threat model, the initial state refers to the optimizer state at the beginning of each local training round, rather than the global initialization at the start of training. This assumption follows existing adaptive FL  protocols that synchronize optimizer states across clients, where the server has access to the optimizer state at the start of each round [6]. Therefore, the attacker has access to $m_{t-1}$ and $v_{t-1}$ as the initial state for the current local round, which is exactly the information required for the derivation in Sec. 3. We will clarify this point in the revised version to avoid ambiguity.
>
> [6] Karimireddy, Sai Praneeth, et al. "Mime: Mimicking centralized stochastic algorithms in federated learning." arXiv preprint arXiv:2008.03606 (2020).

---

> > ### Author Rebuttal · Reviewer_23ff · 2026-04-02
> >
> > Thanks for your clarification. I have a further question regarding the threat model. In practice, the number of local updates can be greater than one. Therefore, do the experiments in this paper only consider the case where the number of local updates is equal to one?

---

> > > ### Author Response · Authors · 2026-04-03
> > >
> > > We thank the reviewer for the question.
> > >
> > > Yes, our current experiments focus on the case where the number of local updates is one.
> > >
> > > This choice follows the dominant setting in existing gradient inversion literature, where most works focus on single-step updates, while achieving stable and high-fidelity reconstruction under multiple local updates remains challenging [7]. Since our primary goal is to investigate gradient inversion beyond SGD in the presence of adaptive optimizers, we adopt this standard setting, which is consistent with prior works [8–10].
> > >
> > > In addition, our method can be extended to multiple local updates. For label recovery, we approximate the accumulated updates as a single Adam update:$\tilde{u} = \frac{\Delta \theta}{\eta \cdot K},$ where $\Delta \theta$ denotes the total parameter update after $K$ local steps and $\eta$ is the local learning rate. This pseudo-update is used for label recovery (Sec. 3.2), while the image reconstruction still matches the true accumulated updates using Eq. 3.
> > >
> > > We find that reconstruction under multiple local updates becomes more effective when the local learning rate is relatively small and when there are no duplicate labels within a batch. We conduct experiments under such a setting with multiple full-batch local steps (batch size = 2, learning rate = 1e-4), with all other configurations following Sec. 4.1. We report the quantitative results in the Table below.
> > >
> > > Reconstruction performance of our method under multiple local update steps.
> > >
> > > | Local Steps | PSNR ↑ | LPIPS ↓ | SSIM ↑ | MSE ↓ |
> > > |---------------|--------|----------|--------|--------|
> > > | 1             | 24.19  | 0.2676   | 0.5940 | 0.0038 |
> > > | 2             | 19.31  | 0.4591   | 0.5064 | 0.0118 |
> > > | 4             | 19.67  | 0.4673   | 0.5158 | 0.0108 |
> > > | 8             | 17.52  | 0.5797   | 0.4718 | 0.0177 |
> > >
> > > As shown, while reconstruction quality degrades compared to the single-step case, our method can still reconstruct images that are visually similar to the original images in this multi-step setting: even with up to 8 local steps, the reconstructed images still achieve a PSNR of 17.52 with respect to the ground truth.
> > >
> > > We regard a more comprehensive study of the multi-step setting as an important direction for future work.
> > >
> > > [7] Carletti, Vincenzo, et al. "SoK: Gradient Inversion Attacks in Federated Learning." 34th USENIX Security Symposium (USENIX Security 25). 2025.
> > > [8] Zhu, Ligeng, Zhijian Liu, and Song Han. "Deep leakage from gradients." Advances in neural information processing systems 32 (2019).
> > > [9] Fang, Hao, et al. "Gifd: A generative gradient inversion method with feature domain optimization." Proceedings of the IEEE/CVF International Conference on Computer Vision. 2023.
> > > [10] Ye, Zipeng, et al. "High-fidelity gradient inversion in distributed learning." Proceedings of the AAAI conference on artificial intelligence. Vol. 38. No. 18. 2024.

---

### Decision · Program_Chairs · 2026-04-30

**Decision:**

Accept (regular)

**Comment:**

This submission extends the previous Gradient Inversion Attacks (GIAs) from SGD to adaptive optimizers, mainly focusing on Adams.

All the reviewers acknowledged its algorithmic contributions to an unexplored scenario. During the rebuttal, most reviewers raised concerns regarding the comprehensiveness of the empirical evaluations and the clarity of novelty. The authors' rebuttal and follow-up comments addressed these concerns well and converged to weak acceptance.

I thus suggest acceptance, and suggest that the authors include the additional empirical results and add the construtive discussions with reviewers to the final version of the paper.